# Diffusion-based learning framework for Constrained Nonconvex Optimization with Weighted Bootstrapped Refinement

**Shutong Ding** [1 2 *]  **Yimiao Zhou** [1 *]  **Ke Hu** [1]  **Xi Yao** [3]  **Junchi Yan** [4]  **Xiaoying Tang** [5]  **Ye Shi** [1 2 †]

## Abstract

Recent advances in diffusion models show promising potential to accelerate nonconvex problem solving by leveraging their multimodality. However, most existing diffusion-based optimization approaches rely on supervised learning and lack a mechanism to enforce constraint satisfaction, which is required in real-world applications. In that case, we investigate and theoretically analyze the inherent problem of supervised diffusion solvers and identify the distributional misalignment problem, i.e., the generated solution distribution often exhibits low probability mass on the feasible region. To resolve this issue, we propose DiOpt, a new diffusion-based learning framework for constrained nonconvex optimization, which effectively learns the mapping from noise to the constraint region. Specifically, this framework operates in two distinct phases: an initial warm-start phase, implemented via supervised learning, followed by a bootstrapping training phase. This dual-phase architecture is designed to iteratively refine solutions, thereby improving the objective function with high constraint satisfaction. Finally, we also employ a solution selection technique in inference for better optimality. Notably, DiOpt is the first successful integration of the diffusion solver in constrained nonconvex optimization. Evaluations on diverse nonconvex tasks demonstrate the superiority of DiOpt in both optimality and constraint satisfaction. Our official page is released at `https://dingsht.tech/diopt-webpage`.

---

[*]Equal contribution [1]ShanghaiTech University [2]MoE Key Laboratory of Intelligent Perception and Human Machine Collaboration [3]China Mobile Communications Company Limited Research Institute [4]Shanghai Jiao Tong University [5]The Chinese University of Hong Kong, Shenzhen. Correspondence to: Ye Shi <shiye@shanghaitech.edu.cn>.

*Proceedings of the 43rd International Conference on Machine Learning*, Seoul, South Korea. PMLR 306, 2026. Copyright 2026 by the author(s).

## 1. Introduction

Nonconvex problems with constraints are fundamental to real-world decision-making systems, spanning critical applications from power grid operations (Pan et al., 2020; Ding et al., 2024b; Cain et al., 2012; Shi et al., 2017) and wireless communications (Du et al., 2024) to robotic motion planning (Li et al., 2025a; Chi et al., 2023). Traditional numerical methods (Nocedal & Wright, 1999; Zimmerman et al., 2011; Wächter & Biegler, 2006) face a fundamental trade-off: either simplify problems through restrictive relaxations (e.g., linear programming approximations) or endure prohibitive computational costs, both unsuitable for safety-critical and real-time systems. Learning-based approaches (Donti et al., 2021; Park & Van Hentenryck, 2023; Zamzam & Baker, 2020; Zhang & Zhang, 2022; Jiang et al., 2024; Zhao & Barati, 2024) emerged as promising alternatives by training neural networks to predict solutions directly, yet they suffer from two critical limitations, as shown in Figure 1: 1) Deterministic one-to-one generation methods exhibit poor performance in handling multi-value mappings (Liang & Chen, 2024). 2) The predicted solutions often have limited overlap with the feasible region, resulting in low feasibility rates.

Recent efforts to address these limitations have turned to diffusion models (Ho et al., 2020; Song et al., 2020a;b), leveraging their multimodal sampling capacity to generate diverse solution candidates (Li et al., 2025a; Pan et al., 2024). Though sampling diversity can reduce single-point failures, existing diffusion-based solvers often still produce low feasibility rates, partly due to a distributional mismatch between their training targets and the feasible set, which becomes more pronounced in high-dimensional problems. Moreover, many approaches rely heavily on supervised training with labeled solution datasets, which can be costly to obtain for complex constrained problems. Some works incorporate additional guidance or projection steps to improve feasibility, leading to slow sampling procedures that scale poorly with dimensionality (Li et al., 2025a; Pan et al., 2024). In addition, some diffusion-based methods attempt to generate constraint-satisfying points through gauge mapping (Li et al., 2025b). However, these approaches typically require a large number of feasible samples during training, which

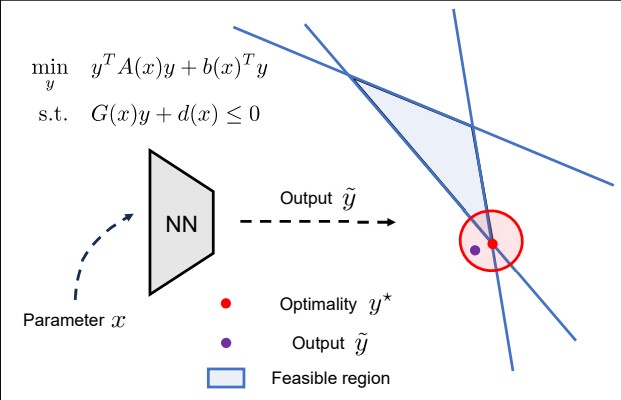

*Figure 1.* A geometric illustration of feasibility challenges in learning-based optimization. The feasible region (blue) and the model-induced output distribution (red) may have limited overlap, especially in high-dimensional settings with multiple constraints. Such distributional misalignment can lead to low feasibility rates in generated solutions.

may be difficult to obtain in many practical settings where feasible solutions are sparse or expensive to generate.

To resolve the absence of an efficient way that employ the powerful diffusion model into nonconvex optimization with constraints, we first analyze the problem of directly applying the supervised diffusion training paradigm in optimization. We then present DiOpt, which operates in two distinct phases: an initial warm-start phase, implemented via supervised learning, followed by a self-supervised phase. In this manner, DiOpt can improve the optimality of solutions in the objective function with high constraint satisfaction. Concretely, DiOpt introduces a target distribution designed to maximize overlap with the constrained region and develops a weighted bootstrapped refinement mechanism to enforce the diffusion solver to approximate this distribution. In addition, a solution selection technique is applied for better optimality and constraint satisfaction during inference. Furthermore, to speed up training, DiOpt maintains a lookup table of historically best candidates, which serves as a more reliable reference set for subsequent updates.

To summarize, our contributions are threefold:

- We reveal and theoretically analyze an inherent limitation in existing diffusion-based approaches for optimization: under standard supervised training, the learned target distribution can be misaligned with the feasible region, which often results in low feasibility rates, especially in high-dimensional settings.

- To mitigate this issue, we propose DiOpt, which augments supervised training with a weighted bootstrapping stage. The bootstrapping mechanism reweights sampled candidates according to constraint violation severity and opti-

mality gap, improving feasibility rates while maintaining competitive solution quality.

- We evaluate DiOpt on diverse constrained nonconvex optimization problems, including synthetic QPSR, synthetic concave QP, Optimal Power Flow and motion retargeting. The results illustrate DiOpt's performance across varied objectives and constraints, and we further analyze its training dynamics and key hyperparameters.

## 2. Related Works

**Learning to Optimize.** To address the high computational cost of classical optimization solvers, Learning to Optimize (L2O) has emerged as a promising approach that leverages machine learning techniques to solve real-world constrained optimization problems. The L2O methods can be generally categorized into two groups: 1) assisting traditional solvers with machine learning techniques to improve their efficiency or performance; 2) approximating the input-output mapping of optimization problems using data-driven models. In the first category, reinforcement learning (RL) has been widely adopted to design better optimization policies for both continuous (Li & Malik, 2016) and discrete decision variables (Liu et al., 2022; Tang et al., 2020). Additionally, neural networks have been used to predict warm-start points for optimization solvers, significantly reducing convergence time (Baker, 2019; Dong et al., 2020). In the second category, deep learning models have been employed to directly approximate solutions for specific problems. For instance, Fioretto et al. (2020); Chatzos et al. (2020) utilize neural networks to solve the OPF problems efficiently. To further improve constraint satisfaction, recent works have integrated advanced techniques into the training process. For example, Donti et al. (2021) introduce gradient-based correction, while Park & Van Hentenryck (2023) incorporate primal-dual optimization methods to ensure the feasibility of the learned solutions.

**Neural Solvers with Hard Constraints.** Despite the challenge of devising general-purpose neural solvers for arbitrary hard constraints, there are also some tailored neural networks (with special layers) for constrained optimization, especially for combinatorial optimization. In these methods, the problem-solving can be efficiently conducted by a single forward pass inference. For instance, in graph matching, or more broadly the quadratic assignment problem, there are a series of works (Wang et al., 2019; Fey et al., 2020) introducing the Sinkhorn layer into the network to enforce the matching constraint. Another example is the cardinality-constrained problem; similar techniques can be devised to ensure the constraints (Brukhim & Globerson, 2018; Wang et al., 2023; Cao & Li, 2024). However, these layers are specifically designed and cannot be used in general settings as addressed in this paper. Moreover, it

*Table 1.* Comparison among existing L2O algorithms across various dimensions.

| Method | Target Task | Real-time Requirement | Low Data Demand | Solution Diversity | Learning Paradigm | Non-differentiable Objective |
|---|---|---|---|---|---|---|
| DC3 (Donti et al., 2021) | continuous, hard | ✓ | ✓ | ✗ | self-supervised | ✗ |
| DiffOPT (Kong et al., 2024) | continuous, soft | ✓ | ✓ | ✓ | supervised | ✓ |
| MBD (Pan et al., 2024) | continuous, soft | ✗ | ✓ | ✓ | N/A | ✓ |
| DiffuSolve (Li et al., 2025a) | continuous, hard | ✓ | ✓ | ✓ | supervised | ✗ |
| RectFlow (Liang & Chen, 2024) | continuous, hard | ✓ | ✗ | ✓ | supervised | ✓ |
| T2T (Li et al., 2024a) | combinatorial, hard | ✗ | ✓ | ✓ | supervised | ✗ |
| DiOpt(*) | continuous, hard | ✓ | ✓ | ✓ | mixed | ✓ |

often requires ground truth for supervision, which cannot be obtained easily in real-world cases.

**Generative Models for Constrained Optimization.** Generative methods for optimization, characterized by sampling from noise, involve models that transform random noise into a distribution over candidate solutions. We briefly review representative lines of work below. One line of work modifies standard diffusion sampling procedures (Zhang et al., 2024; Kurtz & Burdick, 2024; Pan et al., 2024; Kong et al., 2024). These methods reformulate the optimization objective as a problem-induced density and plug it into the sampling dynamics in place of the learned score. However, lacking the high-quality learned proposal, they typically require longer sampling steps to obtain a satisfactory solution. Another line learns an amortized noise-to-solution mapping with neural generators, e.g., CVAE-based (Li et al., 2023) or GAN-based (Salmona et al., 2022) methods. Training-based diffusion approaches include compositional score constructions (Briden et al., 2025) and penalty-based feasibility learning (Li et al., 2025a); related theoretical results are provided in (Liang & Chen, 2024). To enforce the feasibility of generated solutions, PDM (Christopher et al., 2024) introduces projection overhead after each denoising step, while CGD (Kondo et al., 2024) proposes problem-specific post-processing tailored to particular settings. For combinatorial optimization, T2T (Li et al., 2024a) and Fast T2T (Li et al., 2024b) also propose a train-to-test framework.

***Remark.*** However, many training-based approaches are trained in a supervised learning paradigm, which tend to suffer from the distributional misalignment problem illustrated in Figure 1. In contrast, DiOpt uses bootstrapped self-training to learn a generator whose output distribution places substantially more probability mass on near-optimal feasible solutions, without additional labeled-solution data collection. Moreover, DiOpt focuses on constrained nonconvex optimization with continuous (real-valued) variables and generally nonconvex objectives and constraints (e.g., robotics, smart grids, communication systems), which differs from combinatorial optimization (Li et al., 2024a;b; Karalias & Loukas, 2020; Sanokowski et al., 2024; 2025) (e.g., TSP) in both problem structure and methodology. Ta-

ble 1 summarized the differences between our method and several representative L2O approaches in terms of target task, data demand, and learning paradigm.

## 3. Preliminaries

**Learning for Hard-constrained Optimization.** Learning-to-optimize attempts to learn neural networks to generate solutions for a family of optimization problems as

$$\min_{\mathbf{y}} \quad f(\mathbf{y}; \mathbf{x})$$
$$\text{subject to} \quad g_i(\mathbf{y}; \mathbf{x}) \leq 0 \quad i = 1, \cdots, m \qquad (1)$$
$$h_j(\mathbf{y}; \mathbf{x}) = 0 \quad j = 1, \cdots, n$$

where $\mathbf{y} \in \mathbb{R}^{d_y}$ is the decision variable of the optimization problem parameterized by $\mathbf{x} \in \mathbb{R}^{d_x}$. We can use machine learning techniques to learn the mapping from $\boldsymbol{x}$ to its corresponding solution $\mathbf{y}^{\star}$ in an optimization problem family with a similar problem structure. With this mapping, the solution can be calculated faster and more efficiently compared with the classical optimization solver. Moreover, considering the differences between hard equality and inequality constraints, learning-to-optimize methods typically employ different mechanisms to ensure their satisfaction (Donti et al., 2021; Ding et al., 2024b). **Diffusion Models.** Denoising diffusion probabilistic models (DDPM) (Ho et al., 2020) are generative models that create high-quality data by learning to reverse a gradual forward noising process applied to the training data. Given a dataset $\{\mathbf{y}_0^i\}_{i=1}^D$ for $\mathbf{y}_0^i \sim q(\mathbf{y}_0)$, the forward process $\{\mathbf{y}_{0:T}\}$ adds Gaussian noise to the data with pre-defined schedule $\{\beta_{1:T}\}$:

$$q(\mathbf{y}_t \mid \mathbf{y}_{t-1}) := \mathcal{N}(\mathbf{y}_t; \sqrt{1-\beta_t}\mathbf{y}_{t-1}, \beta_t \mathbf{I}). \qquad (2)$$

Using the Markov chain property, we can obtain the analytic marginal distribution of conditioned on $\boldsymbol{x}_0$:

$$q(\mathbf{y}_t \mid \mathbf{y}_0) = \mathcal{N}(\mathbf{y}_t; \sqrt{\bar{\alpha}_t}\mathbf{y}_0, (1-\bar{\alpha}_t)\mathbf{I}), \forall t \in \{1, \ldots, T\}, \qquad (3)$$

where $\alpha_t = 1 - \beta_t$ and $\bar{\alpha}_t = \prod_{s=0}^T \alpha_s$. Given $\boldsymbol{x}_0$, it's easy to obtain a noisy sample by the reparameterization trick.

$$\mathbf{y}_t = \sqrt{\bar{\alpha}_t}\mathbf{y}_0 + \sqrt{1-\bar{\alpha}_t}\epsilon, \epsilon \in \mathcal{N}(\mathbf{0}, \boldsymbol{I}). \qquad (4)$$

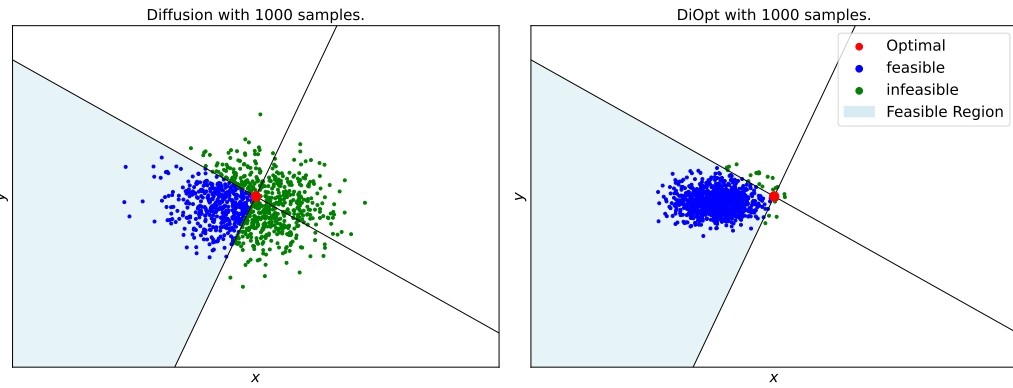

*(a)* Diffusion solver trained in a supervised manner.    *(b)* Diffusion solver trained by DiOpt.

*Figure 2.* **Comparison between supervised diffusion and DiOpt on a toy example.** It can be observed that the distribution of diffusion trained in a supervised manner (a) approximates a Gaussian distribution centered around the optimal point, which leads to a low feasibility rate. For detailed settings of this toy example, please refer to Appendix G.2 for more details.

DDPMs reverse the diffusion process via a parameterized distribution $p_\theta(\boldsymbol{y}_{t-1} \mid \boldsymbol{y}_t)$. In practice, we train a function approximator $\boldsymbol{\epsilon}_\theta(\boldsymbol{y}_t, t)$ to predict the added noise $\epsilon \sim \mathcal{N}(\boldsymbol{0}, \boldsymbol{I})$. The resulting simplified loss is:

$$\mathbb{E}_{t \sim [1,T], \mathbf{y}_0, \boldsymbol{\epsilon}_t} \left[ ||\boldsymbol{\epsilon}_t - \boldsymbol{\epsilon}_\theta(\sqrt{\bar{\alpha}_t}\mathbf{y}_0 + \sqrt{1 - \bar{\alpha}_t}\boldsymbol{\epsilon}_t, t)||^2 \right]. \tag{5}$$

After training, DDPM generates samples according to

$$\boldsymbol{y}_{t-1} = \frac{1}{\sqrt{\alpha_t}} \left( \boldsymbol{y}_t - \frac{1 - \alpha_t}{\sqrt{1 - \bar{\alpha}_t}} \boldsymbol{\epsilon}_\theta(\boldsymbol{y}_t, t) + \eta \cdot \sigma_t \boldsymbol{z} \right),$$
$$\boldsymbol{z} \sim \mathcal{N}(\boldsymbol{0}, \mathbf{I}), \quad t = T, ..., 1 \tag{6}$$

where $\eta$ denotes the noise level.

Owing to the powerful expressiveness of diffusion, recent works on online reinforcement learning (Yang et al., 2023; Ding et al., 2024a; Psenka et al., 2023) begin to pay more attention on the diffusion model. One representative work is QVPO (Ding et al., 2024a), which employ the diffusion variational loss weighted by the truncated advantage to train the diffusion model:

$$\mathbb{E}_{t, \boldsymbol{a}, \boldsymbol{\epsilon}_t} \left[ w(\boldsymbol{s}, \boldsymbol{a}) \left\| \boldsymbol{\epsilon}_t - \boldsymbol{\epsilon}_\theta \left( \sqrt{\bar{\alpha}_t} \boldsymbol{a} + \sqrt{1 - \bar{\alpha}_t} \boldsymbol{\epsilon}_t, t \right) \right\|^2 \right], \tag{7}$$

where $w(\boldsymbol{s}, \boldsymbol{a}) = \max(0, A(\boldsymbol{s}, \boldsymbol{a}))$. This weighted mechanism has significantly inspired the design bootstrapped refinement procedure in DiOpt.

***Remark.*** However, QVPO is specifically designed for reinforcement learning, and its weighting scheme and training paradigm cannot be directly applied to nonconvex optimization problems. In nonconvex optimization, the design of weighting functions must carefully balance constraint satisfaction and optimality. Moreover, due to the high nonconvexity of the objective or constraints, obtaining a diffusion solver within a limited training budget remains a significant

challenge. Hence, Our DiOpt is proposed to resolve these challenges as we detail it in Section 5.

## 4. Why Supervised Diffusion Tends to Generate Infeasible Solutions

As mentioned in Section 1, how to apply diffusion models to constrained continuous optimization problems effectively and then generate near-optimal solutions without violating the complex (nonconvex) constraints remains largely unexplored. To investigate the underlying reason, we conduct a toy example, which applies diffusion models trained in a supervised paradigm to a two-dimensional QP problem with linear constraints. Figure 2 shows that this kind of diffusion model tends to generate points around the optimal solution, and the distribution is an approximate Gaussian distribution. Nevertheless, the feasible region only has a small overlap with the Gaussian ball, which leads to the diffusion solver violating the constraints with a high probability (i.e., the number of blue points (feasible) is greater than that of the green points).

Based on this finding, we also present a theoretical proof, as Theorem 1, that in high-dimensional optimization problems, exemplified by linear programming, diffusion models trained via supervised learning become increasingly prone to constraint violations, with the violation probability growing exponentially towards one as the dimensionality increases.

**Theorem 1.** *(**Feasibility in Linear Programming.**) Given a d-dimensional linear programming problem of the form:*

$$\min_{y \in \mathbb{R}^{d_y}} c^T y \quad subject \ to \quad a_i^T y \leq b_i, \quad i = 1, \cdots, N,$$

*where $a_i$ represents a unit normal vector, drawn independently and uniformly from the unit sphere $S^{d_y - 1}$. Let $y^\star$ be the unique solution to this linear programming*

**Algorithm 1** Bootstraping-based training process of DiOpt

---

**Input:** training dataset $X$, the objective function $f(\boldsymbol{y}; \boldsymbol{x})$ and constraints $h(\boldsymbol{y}; \boldsymbol{x}), g(\boldsymbol{y}; \boldsymbol{x})$, the noise network $\epsilon_\theta$ of diffusion model $\mathcal{D}_\theta$, look-up table $\mathcal{B}$, Number of Training Epochs $N_e$, Supervised Ratio $r_s$, Number of Training Samples $K_t$.

**for** $n = 0$ **to** $N_e - 1$ **do**
  **if** $n \leq \lfloor r_s \cdot N_e \rfloor$ **then**
    Train $\mathcal{D}_\theta$ by (8)
    Continue
  **end if**
  **if** n **mod** $2 = 0$ **then**
    Reset the weight function as (11)
  **else**
    Reset the weight function as (12)
    // reset diffusion with feasible points
  **end if**
  **for** $\boldsymbol{x}_i$ in $X$ **do**
    $\boldsymbol{y}_1, \ldots, \boldsymbol{y}_{K_t} \overset{i.i.d}{\sim} \mathcal{D}_\theta(\boldsymbol{x}_i)$
    $\boldsymbol{y}_{\text{best}} \leftarrow \mathcal{B}(\boldsymbol{x}_i)$
    Endow $\boldsymbol{y}_1, \ldots, \boldsymbol{y}_{K_t}, \boldsymbol{y}_{\text{best}}$ with weights according to (13)
    Train $\mathcal{D}_\theta$ by (14)
    Update $\mathcal{B}(\boldsymbol{x}_i)$ with $\boldsymbol{y}_1, \ldots, \boldsymbol{y}_{K_t}$ by (15)
  **end for**
**end for**
**Return** $\mathcal{D}_\theta$

---

*problem, and we define a neighborhood ball $B_\epsilon(y^\star) = \{y : \|y - y^\star\| \leq \epsilon\}$. For sufficiently small $\epsilon > 0$, there is an asymptotic bound on the probability that a point uniformly sampled from $B_\epsilon(y^\star)$ lies in the feasible region $\mathcal{C}$.*

$$\mathbb{P}_{x \sim B_\epsilon(y^\star)}\left(y \in \mathcal{C}\right) \approx \frac{1}{2^{d_y}}.$$

For more general nonlinear inequality constraints, one can similarly linearize them within an infinitesimal neighborhood and reduce to the case described in this theorem. This theorem also confirms our findings in the toy example in Figure 2. The proof leverages results from stochastic geometry (Cover & Efron, 1967; WENDEL, 1962), with details provided in Appendix A. Besides, we also provide a theoretical analysis to illustrate this issue cannot be resolved via multiple sampling in Appendix B. Hence, it suggests that training diffusion models with optimal samples is not an ideal choice for constrained optimization problems, and a more effective method needs to be developed to better incorporate the diffusion model into constrained optimization.

## 5. Methodology

To address the issue mentioned in Section 4, we propose DiOpt, a self-supervised **Di**ffusion-based learning framework for Constrained **Opt**imization in this section. As shown in Figure 3, DiOpt trains the diffusion model in a bootstrapping mechanism via weighted variational loss of diffusion and applies the candidate *solution selection* technique during the evaluation stage to further boost the solution quality. The training procedure of DiOpt is presented in Algorithm 1.

**Target Distribution for Diffusion Training.** To apply the diffusion model to optimization problems, we train it using the following objective:

$$\mathbb{E}_{\boldsymbol{x}, \boldsymbol{y}^\star, \epsilon, t}\left[\|\epsilon - \epsilon_\theta(\boldsymbol{y}_t, \boldsymbol{x}, t)\|_2^2\right] \qquad (8)$$

where $\boldsymbol{y}^\star \in \mathbb{R}^{d_y}$ denotes the optimal solution corresponding to $\boldsymbol{x} \in \mathbb{R}^{d_x}$ and $\epsilon_\theta$ is the noise network of diffusion model $\mathcal{D}_\theta$. The near-optimal distribution learned by the diffusion model trained in this manner is ultimately similar to that shown in Figure 1. However, in high-dimensional settings, the intersection between the near-optimal region of the diffusion model and the feasible region becomes too small, which leads to infeasibility issues when applying the diffusion model to optimization problems. In that case, it is necessary to define another target distribution for diffusion models to enforce their constraint satisfaction. Referring to (Liang & Chen, 2024), we define the target distribution that corresponds to the near-optimal feasible region as

$$p(\boldsymbol{y}; \boldsymbol{x}) \sim \mathbb{I}_{\mathcal{C}(\boldsymbol{x})}(\boldsymbol{y}) \exp\left(-\beta f(\boldsymbol{y}; \boldsymbol{x})\right), \qquad (9)$$

where $\boldsymbol{y} \sim \mathcal{D}_\theta(\boldsymbol{x})$, $\mathcal{C}(x)$ indicates the constraint region that satisfies $g_i(\boldsymbol{y}; \boldsymbol{x}) \leq 0, h_j(\boldsymbol{y}; \boldsymbol{x}) = 0$ and $\mathbb{I}_{\mathcal{C}(\boldsymbol{x})}$ is the indicator function that judges the satisfaction of the constraint.

**Training Diffusion with Bootstrapping.** While Liang & Chen (2024) has argued that it is essential to approximate the above target distribution rather than merely an optimal point, it is difficult to sample from Eq. (9) efficiently. Therefore, constructing a dataset based on the target distribution for training diffusion models is also impossible. To avoid this problem, we divert our attention to self-supervised learning. Motivated by (Ding et al., 2024a), we design a diffusion-based learning framework for constrained optimization, which implements diffusion training in a bootstrapping manner. To better adapt the weighted training mechanism to constrained optimization, we design entirely different weighting functions to address both constraint satisfaction and objective optimality. In addition, we introduce techniques such as a reset operation and a look-up table for higher feasibility and faster convergence. In this way, DiOpt can naturally converge to the target distribution without manufacturing a corresponding dataset.

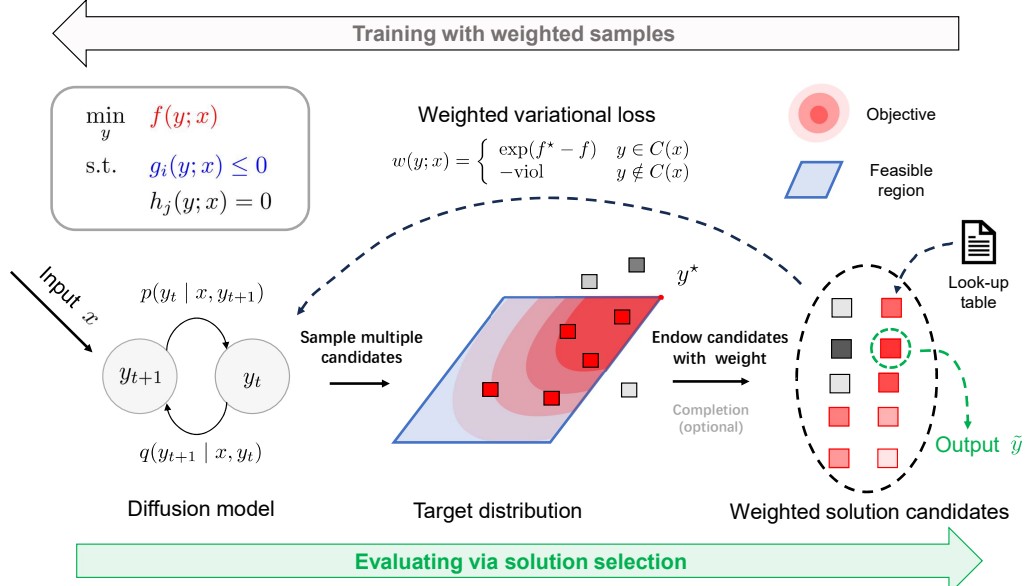

*Figure 3.* Training and evaluating procedure of DiOpt.

Concretely, we try to utilize the parallelism of the diffusion model and generate a certain number of candidate points for one specific problem, and then endow the candidate points with weights related to the constraint violation and objective value. The diffusion model will be trained with these weighted candidate points according to:

$$\mathcal{L}(\theta) := \mathbb{E}_{\boldsymbol{x},\boldsymbol{y},\boldsymbol{\epsilon},t} \left[ \omega(\boldsymbol{y};\boldsymbol{x}) \left\| \boldsymbol{\epsilon} - \boldsymbol{\epsilon}_\theta \left(\boldsymbol{y}_t, \boldsymbol{x}, t\right) \right\|_2^2 \right], \quad (10)$$

Here, the weight can be viewed as the importance of training points. Hence, the diffusion model can approximately converge to the distribution defined by the weight function after enough iterations. Thus, the weight function design is essential to ensure that the diffusion model to converge to the target distribution. Based on that, we classify points into two cases and design two different weight functions for them:

$$\omega(\boldsymbol{y};\boldsymbol{x}) := \begin{cases} \exp\left(f^\star(\boldsymbol{x}) - f(\boldsymbol{y};\boldsymbol{x})\right) & y \in \mathcal{C}(\boldsymbol{x}) \\ -\sum_i \max(g_i(\boldsymbol{y};\boldsymbol{x}),0) & y \notin \mathcal{C}(\boldsymbol{x}) \end{cases}, \quad (11)$$

where $f^\star(\boldsymbol{x})$ indicates the optimal objective value of the optimization problem with parameter $\boldsymbol{x}$. One of the principal ideas for this weight function is that all the feasible points have positive weights and the infeasible points have negative weights. In that case, the diffusion model will converge to the feasible region and then consider the optimality of the points inside the constraint region. Besides, it is worth noting that $f^\star(\boldsymbol{x})$ can be replaced with the estimated lower bound of the objective function. This term actually avoids the numerical explosion of the exponential function.

Furthermore, to ensure that the trained model prioritizes feasibility over optimality, we reset the weight function every training epoch as follows:

$$\omega(\mathbf{y};\mathbf{x}) = -\sum_i \max(g_i(\mathbf{y};\mathbf{x}),0) \quad (12)$$

***Remark.*** If we continue updating diffusion with the weight of Eq. (11), the output of the diffusion solver will resemble that of supervised learning and will be prone to generating infeasible points. The detailed proof can be referred to in the Appendix C. Hence, this formulation penalizes constraint violations ($g_i(\mathbf{y};\mathbf{x}) > 0$), systematically steering the model toward the feasible region.

However, there is still a problem to be resolved in our weight function. As illustrated in (Ding et al., 2024a), the weight in Eq. (10) must always be positive. Hence, we perform a modification on the final weight when there exists a candidate point with a negative weight.

$$\widetilde{\omega}(\boldsymbol{y};\boldsymbol{x}) = \max\left(\omega(\boldsymbol{y};\boldsymbol{x}) - \bar{\omega}, 0\right),$$

$$\text{where } \bar{\omega} = \frac{1}{K_t}\sum_{i=1}^{K_t} \omega(\boldsymbol{y}_i;\boldsymbol{x}), \quad \boldsymbol{y}_i \overset{i.i.d}{\sim} \mathcal{D}_\theta(\boldsymbol{x}) \quad (13)$$

As shown in (Ding et al., 2024a), $\widetilde{\omega}$ is equivalent to $\omega$ for diffusion training, we can ensure the diffusion model converges to the target distribution with the modified weight $\widetilde{\omega}$. This leads to the following approach:

$$\mathbb{E}_{\boldsymbol{x},\widetilde{\boldsymbol{y}},\epsilon,t} \left[ \widetilde{\omega}(\widetilde{\boldsymbol{y}},\boldsymbol{x}) \left\| \epsilon - \epsilon_\theta(\widetilde{\boldsymbol{y}}_t,\boldsymbol{x},t) \right\|_2^2 \right],$$

$$\widetilde{\boldsymbol{y}} = \underset{\boldsymbol{y} \in \{\boldsymbol{y}_1,\ldots,\boldsymbol{y}_{K_t}\}}{\arg\max} \omega(\boldsymbol{y};\boldsymbol{x}) \quad (14)$$

Due to the curse of dimensionality (Bellman et al., 1957), training the model purely based on bootstrapping may make it difficult for the diffusion model to explore sufficiently good solutions. Therefore, at the initial stage of training, we utilize Eq. (8) to train the diffusion model and guide it towards the vicinity of the optimal point. Afterwards, Eq. (14) is employed to further train the model, ensuring feasibility. Throughout all training epochs, the proportion of supervised learning is denoted as $r_s$.

**Handling of Equality Constraint.** We apply a standard technique to handling hard equality constraints in learn-to-optimize literature (Donti et al., 2021; Ding et al., 2024b). Consider decision variables $y \in \mathbb{R}^n$ subject to $m$ equality constraints $h_i(y; x) = 0$ for $i = 1, \ldots, m$. We partition $y$ into basic and nonbasic variables, $y = (y_b, y_n)$, where $y_b \in \mathbb{R}^{n-m}$ parameterizes the degrees of freedom and $y_n \in \mathbb{R}^m$ is determined by the equalities.

Given $y_b$, we solve $h_i(y_b, y_n; x) = 0$ with an equation solver to obtain $y_n$, which enforces the equalities exactly. Accordingly, for problems with equality constraints, DiOpt's diffusion model generates only $y_b$, and we recover $y_n$ via Newton Method.

Compared with existing approaches that often rely on costly iterative updates to handle hard inequality constraints while preserving equalities, DiOpt focuses on improving feasibility under hard inequality constraints

**Quality Assurance Mechanism.** Due to the inherent randomness of diffusion model, we cannot guarantee that each round of sampling yields a better solution than the previous one. To ensure training stability, we maintain a look-up table $\mathcal{B} : \mathbb{R}^{d_x} \to \mathbb{R}^{d_y}$ that stores the best solution $y$ found so far for each input.

Let $y_{\text{best}}$ denote the optimal solution stored in $\mathcal{B}$ for a given problem parameter $x$. The update rule for $\mathcal{B}$ is formulated as:

$$\mathcal{B}(\mathbf{x}) = \underset{y \in \{y_1, \ldots, y_{K_t}, y_{\text{best}}\}}{\arg \max} \omega(y; x) \tag{15}$$

After training, we employ *Solution Selection* for inference:

$$\widetilde{\mathbf{y}} = \underset{\mathbf{y} \in \{\mathbf{y}_1, \ldots, \mathbf{y}_{K_e}\}}{\arg \max} \omega(\mathbf{y}; \mathbf{x}), \quad \mathbf{y}_1, \ldots, \mathbf{y}_{K_e} \overset{i.i.d}{\sim} \mathcal{D}_\theta(\mathbf{x}) \tag{16}$$

By selecting a sufficiently large $K_e$, we ensure a high probability of obtaining a near-optimal solution.

# 6. Experiment

Building upon the illustrative Toy Example demonstration in Figure 2b, we now conduct a comprehensive empirical evaluation of DiOpt on five complex and challenging nonconvex optimization tasks. These tasks include two manually constructed problems: QPSR and Concave QP (CQP), as well as challenging real-world benchmarks such as AC Optimal Power Flow (ACOPF), a non-convex optimization problem in power systems, and Motion Retargeting (He et al., 2024), which maps human motion to a humanoid robot under kinematic constraints. CQP is a specially designed benchmark. Because of its distinctive objective function, it is more prone to infeasibility compared with other tasks. Spanning diverse domains, these problems highlight DiOpt's versatility and representation capability. Detailed experimental settings, ablation results, benchmark formulations and baseline settings are provided in Appendix D, F, G and Appendix H.

## 6.1. Comparative Evaluation

We use IPOPT (Wächter & Biegler, 2006) as a solver reference. Boldface indicates the best value in each column, and "N/A" denotes invalid runs. Objectives marked with $\times$ are unreliable due to low feasibility (Feasibility < 50%), while $\checkmark$ indicates Feasibility $\geq 50\%$. An inequality constraint $g_i(y; x)$ is considered violated if $g_i(y; x) > \epsilon$; we report the mean/max violation and the number of violated constraints, with $\epsilon = 0.01$. We utilize the purely supervised stage of DiOpt as a proxy for RectFlow (Liang & Chen, 2024), given that they share an identical training paradigm and differ solely in the generative backbone. Detailed analysis and justification are provided in Appendix H.4. Accordingly, we denote this baseline as RectFlow in our Table 2.

DiOpt achieves the best trade-off between feasibility and optimality gap on most tasks, except ACOPF57, validating the benefit of our bootstrapped refinement. Equality solver causes MBD to fail, but it is hard to satisfy equality constraints without it. DC3 exhibits numerical instability on the CQP benchmark. This issue arises because CQP is a specially designed and challenging benchmark; detailed analysis is provided in Appendix G. RectFlow baselines can attain small gaps but often collapse in feasibility, remaining near zero on most tasks even with $K_e = 64$; see Appendix G for details.

## 6.2. Ablation Study on Reset Operation

In this section, we conduct a simple ablation study to illustrate the effect of the reset mechanism introduced in Section 5. Without this mechanism, once DiOpt discovers a point that is extremely close to the optimum, the subsequent weighted selection tends to repeatedly choose this single point. This behavior can cause DiOpt to degenerate toward supervised diffusion to some extent. The reset step prevents such collapse by steering the method toward the feasible region rather than a specific solution, thereby preserving feasibility. We evaluate this effect on the QPSR task, and the results are reported in Table 3. As shown, incorporating the reset mechanism leads to substantially improved feasibility. A more detailed theoretical explanation, along with

*Table 2.* Performance comparison (mean ± std. dev.). DiOpt uses $K_e = 64$ and DC3 uses 200 correction steps. The light blue row indicates the reference solver (IPOPT). All learning-based methods are trained on the training set and evaluated on a separate test set.

| Problem | Method | Feasibility(%)↑ | Gap(%)↓ | Objective↓ | Time(s)↓ | Ineq Mean↓ | Ineq Max↓ | Ineq Num Viol↓ |
|---|---|---|---|---|---|---|---|---|
| QPSR | IPOPT | $100.00\% \pm 0.00$ | $0.00\% \pm 0.00$ | $-9.77 \pm 0.42$✓ | $0.62 \pm 0.10$ | $0.00 \pm 0.00$ | $0.00 \pm 0.00$ | $0.00 \pm 0.00$ |
| | RectFlow | $0.00\% \pm 0.00$ | $0.43\% \pm 2.87$ | $-9.73 \pm 0.51$× | $0.01 \pm 0.00$ | $0.01 \pm 0.02$ | $0.34 \pm 0.48$ | $11.84 \pm 2.82$ |
| | DC3 | $20.65\% \pm 0.00$ | $33.61\% \pm 7.00$ | $-6.49 \pm 0.78$× | $0.01 \pm 0.00$ | $0.00 \pm 0.01$ | $0.48 \pm 0.64$ | $2.03 \pm 2.18$ |
| | MBD | $0.04\% \pm 0.00$ | $4101.15\% \pm 4377.28$ | $390.88 \pm 428.73$× | $0.01 \pm 0.00$ | $11.00 \pm 7.84$ | $83.11 \pm 53.43$ | $95.84 \pm 19.66$ |
| | DiOpt* | $\mathbf{81.87\% \pm 0.00}$ | $\mathbf{2.48\% \pm 0.78}$ | $-9.53 \pm 0.43$✓ | $0.01 \pm 0.00$ | $0.00 \pm 0.00$ | $\mathbf{0.01 \pm 0.04}$ | $\mathbf{0.23 \pm 0.47}$ |
| CQP | IPOPT | $100.00\% \pm 0.00$ | $0.00\% \pm 0.89$ | $-37.18 \pm 0.70$✓ | $3.22 \pm 1.02$ | $0.00 \pm 0.00$ | $0.00 \pm 0.00$ | $0.00 \pm 0.00$ |
| | RectFlow | $0.00\% \pm 0.00$ | $1.47\% \pm 0.85$ | $-36.68 \pm 0.71$× | $0.01 \pm 0.00$ | $0.00 \pm 0.00$ | $0.33 \pm 0.18$ | $11.27 \pm 1.90$ |
| | DC3 | $0.00\% \pm 0.00$ | $N/A$ | $N/A$× | $0.01 \pm 0.00$ | $N/A$ | $N/A$ | $N/A$ |
| | MBD | $0.00\% \pm 0.00$ | $45672.14\% \pm 23099.09$ | $-17005.74 \pm 8566.45$× | $0.01 \pm 0.00$ | $23.51 \pm 8.42$ | $602.93 \pm 214.30$ | $70.64 \pm 5.15$ |
| | DiOpt* | $\mathbf{69.95\% \pm 0.00}$ | $7.04\% \pm 1.36$ | $-34.57 \pm 0.64$✓ | $0.01 \pm 0.00$ | $0.00 \pm 0.00$ | $\mathbf{0.05 \pm 0.22}$ | $\mathbf{0.75 \pm 1.82}$ |
| RETARGETING | IPOPT | $100.00\% \pm 0.00$ | $0.05\% \pm 1.22$ | $1.73 \pm 0.51$✓ | $4.36 \pm 21.500$ | $0.00 \pm 0.00$ | $0.00 \pm 0.00$ | $0.00 \pm 0.00$ |
| | RectFlow | $100.00\% \pm 0.00$ | $1.24\% \pm 2.52$ | $1.74 \pm 0.50$✓ | $0.01 \pm 0.00$ | $0.00 \pm 0.00$ | $0.00 \pm 0.00$ | $0.00 \pm 0.00$ |
| | DC3 | $95.86\% \pm 0.00$ | $30.16\% \pm 37.68$ | $2.14 \pm 0.53$× | $0.00 \pm 0.00$ | $0.00 \pm 0.00$ | $0.01 \pm 0.06$ | $0.04 \pm 0.20$ |
| | MBD | $0.00\% \pm 0.00$ | $169.20\% \pm 57.09$ | $4.49 \pm 1.08$× | $0.00 \pm 0.00$ | $0.06 \pm 0.01$ | $2.30 \pm 0.33$ | $1.00 \pm 0.00$ |
| | DiOpt* | $\mathbf{100.00\% \pm 0.00}$ | $\mathbf{0.65\% \pm 1.19}$ | $1.74 \pm 0.51$✓ | $0.01 \pm 0.00$ | $0.00 \pm 0.00$ | $\mathbf{0.00 \pm 0.00}$ | $\mathbf{0.00 \pm 0.00}$ |
| ACOPF57 | IPOPT | $100.00\% \pm 0.00$ | $0.00\% \pm 0.00$ | $3.81 \pm 0.64$✓ | $0.42 \pm 0.04$ | $0.00 \pm 0.00$ | $0.00 \pm 0.00$ | $0.00 \pm 0.00$ |
| | RectFlow | $81.33\% \pm 0.00$ | $\mathbf{0.19\% \pm 0.71}$ | $3.81 \pm 0.64$✓ | $0.02 \pm 0.00$ | $0.00 \pm 0.00$ | $0.01 \pm 0.01$ | $0.26 \pm 0.56$ |
| | DC3 | $\mathbf{94.00\% \pm 0.00}$ | $0.40\% \pm 0.85$ | $3.82 \pm 0.63$✓ | $0.01 \pm 0.00$ | $0.00 \pm 0.00$ | $\mathbf{0.00 \pm 0.00}$ | $\mathbf{0.07 \pm 0.29}$ |
| | MBD | $0.00\% \pm 0.00$ | $22.36\% \pm 14.01$ | $4.74 \pm 1.31$× | $1.28 \pm 0.00$ | $0.02 \pm 0.01$ | $1.30 \pm 0.54$ | $4.30 \pm 1.28$ |
| | DiOpt* | $93.33\% \pm 0.00$ | $0.24\% \pm 0.91$ | $3.81 \pm 0.64$✓ | $0.02 \pm 0.00$ | $0.00 \pm 0.00$ | $0.00 \pm 0.01$ | $0.09 \pm 0.33$ |
| ACOPF118 | IPOPT | $100.00\% \pm 0.00$ | $0.00\% \pm 0.00$ | $13.11 \pm 1.22$✓ | $0.97 \pm 0.042$ | $0.00 \pm 0.00$ | $0.00 \pm 0.00$ | $0.00 \pm 0.00$ |
| | RectFlow | $0.00\% \pm 0.00$ | $2.90\% \pm 0.21$ | $13.49 \pm 1.25$× | $0.07 \pm 0.00$ | $0.00 \pm 0.00$ | $0.38 \pm 0.14$ | $9.10 \pm 1.30$ |
| | DC3 | $43.00\% \pm 0.00$ | $2.49\% \pm 0.19$ | $13.44 \pm 1.24$× | $0.06 \pm 0.00$ | $0.00 \pm 0.00$ | $0.02 \pm 0.03$ | $0.73 \pm 0.76$ |
| | MBD | $0.00\% \pm 0.00$ | $37.39\% \pm 12.81$ | $17.86 \pm 0.33$× | $5.34 \pm 0.00$ | $0.03 \pm 0.01$ | $4.50 \pm 2.41$ | $23.16 \pm 1.33$ |
| | DiOpt* | $\mathbf{84.33\% \pm 0.00}$ | $\mathbf{2.26\% \pm 0.11}$ | $13.41 \pm 1.23$✓ | $0.07 \pm 0.00$ | $0.00 \pm 0.00$ | $\mathbf{0.01 \pm 0.01}$ | $\mathbf{0.20 \pm 0.40}$ |

*Table 3.* Effect of the reset mechanism on feasibility in the QPSR task.

| Metric | Without Reset | With Reset |
|---|---|---|
| Feasibility (%) ↑ | 70.03 | **81.19** |

additional experiments on broader benchmarks, is provided in Appendix C.

### 6.3. Train procedure

This section illustrates the training dynamics of DiOpt and RectFlow as observed on the ACOPF118 and QPSR tasks. For training dynamics results across other tasks, please refer to Appendix F.5. As shown in Figure 4, once bootstrapping begins (**2000** epochs), DiOpt immediately exhibits a significant reduction in constraint violation metrics.

Meanwhile, the objective-related metrics initially experience a slight increase, followed by a gradual decrease toward near-optimal values. This behavior indicates that, after bootstrapping is activated, DiOpt first guides the sampling distribution into the interior of the feasible region, and subsequently steers it progressively toward the neighborhood of the optimal solution. In this way, DiOpt achieves feasibility improvement while preserving near-optimality.

### 6.4. Training Time

We report the training runtime of DiOpt under different time steps $T$ across all tasks in Table 4. For the ACOPF family of problems, a dominant fraction of the overall runtime is spent on completing the network-generated solutions to satisfy

*Table 4.* Training time (hours) of DiOpt across different time steps on each problem.

| Problem | $T = 5$ | $T = 10$ | $T = 20$ | $T = 100$ |
|---|---|---|---|---|
| ACOPF-118 | 34.36 | 34.84 | 35.54 | 42.62 |
| ACOPF-57 | 7.49 | 7.85 | 8.51 | 14.36 |
| QPSR | 3.09 | 5.73 | 11.03 | 53.35 |
| CQP | 3.08 | 5.75 | 11.06 | 53.41 |
| Retargeting | 0.74 | 1.19 | 2.08 | 9.98 |

physical constraints; consequently, increasing the number of time steps $T$ does not introduce a significant increase in training time. By contrast, for QPSR, CQP, and Retargeting, the overhead of completing infeasible solutions is relatively small, and the primary computational cost lies in the iterative forward passes of the network, which scale roughly linearly with $T$. As a result, using a larger number of time steps leads to a substantially higher training cost for these tasks.

## 7. Conclusion and Outlook

This work first explored diffusion models for constrained optimization, revealing that purely supervised methods struggle with feasibility as constraint dimensionality increases. To address this, we proposed DiOpt, a diffusion training framework combining supervised and self-supervised learning. DiOpt guides sampling towards feasibility using a target distribution and improves optimality via weighted self-supervised training. Extensive experiments validate DiOpt's effectiveness, demonstrating improved feasibility and preserved objective quality compared to baselines. Despite its strengths, DiOpt faces limitations in training efficiency

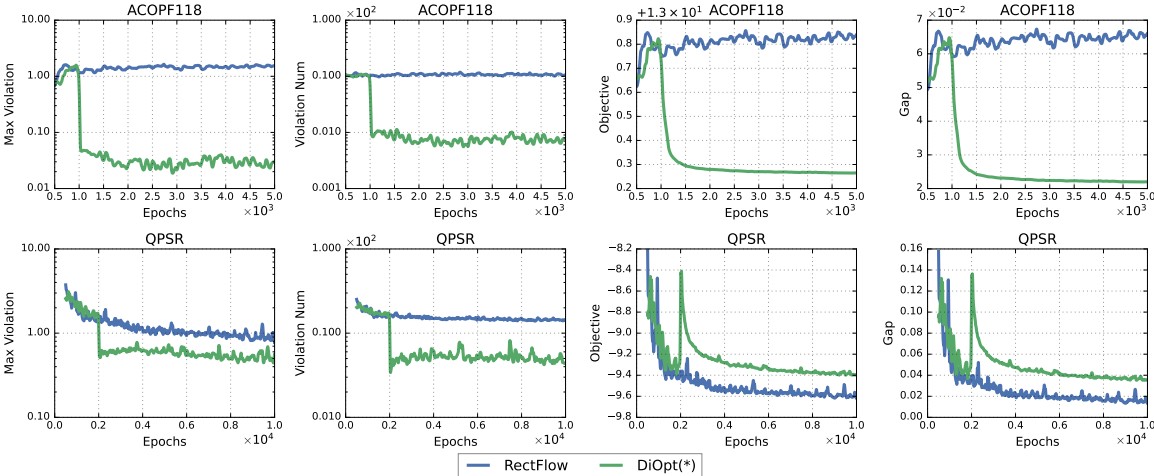

*Figure 4.* Training dynamics on ACOPF118 (top) and QPSR (bottom) tasks. DiOpt (green) demonstrates a sharp reduction in constraint violations at epoch 2000 (1000 for ACOPF118) upon initiating bootstrapping. This mechanism triggers a transient perturbation in the objective and optimality gap, which subsequently converges to near-optimal levels.

due to its self-supervised nature and the costly feasibility completion needed for equality constraints.

## Acknowledgment

This work was supported by the National Natural Science Foundation of China (62303319, 62406195), HPC Platform of ShanghaiTech University, and MoE Key Laboratory of Intelligent Perception and Human-Machine Collaboration (ShanghaiTech University), Shanghai Engineering Research Center of Intelligent Vision and Imaging. This work was also supported in part by computational resources provided by Fcloud CO., LTD.

## Impact Statement

This paper addresses how to solve the nonconvex problems via a diffusion model in a machine learning paradigm, especially for constrained problems. We believe our technology can enhance the application of AI in more real-world scenarios with safety requirements.

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

# A. Proof for Theorem 1

**Theorem 1** *(Feasibility in Linear Programming.) Given a d-dimensional linear programming problem of the form:*

$$\min_{y \in \mathbb{R}^{d_y}} c^T y \quad subject\ to \quad a_i^T y \leq b_i, \quad i = 1, \cdots, N,$$

*where $a_i$ represents a unit normal vector, drawn independently and uniformly from the unit sphere $S^{d_y-1}$. Let $y^\star$ be the unique solution to this linear programming problem, and we define a neighborhood ball $B_\epsilon(y^\star) = \{y : \|y - y^\star\| \leq \epsilon\}$. For sufficiently small $\epsilon > 0$, there is an asymptotic bound on the probability that a point uniformly sampled from $B_\epsilon(y^\star)$ lies in the feasible region $\mathcal{C}$.*

$$\mathbb{P}_{y \sim B_\epsilon(y^\star)} (y \in \mathcal{C}) \approx \frac{1}{2^d}.$$

*Proof.* Motivated by (Cover & Efron, 1967), we can find that, for each linear equation $a_i^T y = 0$, it defines a hyperplane that divides $\mathbb{R}^d$ into two half-spaces, i.e.,

$$A_i := \{y : \operatorname{sgn}(a_i^T y) = 1\}, -A_i := \{y : \operatorname{sgn}(a_i^T y) = -1\}. \tag{17}$$

In that case, we can observe that $d_y$ linearly independent linear equations will partition the n-dimensional space $\mathbb{R}^{d_y}$ into $2^{d_y}$ distinct regions, intersecting at the origin. In other words, let $\delta_i = \operatorname{sgn}(a_i^T y)$ and then we have $2^{d_y}$ choices for $\{\delta_1, \cdots, \delta_{d_y}\} = \{\pm 1, \cdots, \pm 1\}$. Considering the equivalence of each region, a random point sampled uniformly from the unit ball in the n-dimensional space falls into any particular region with an expected probability of $\frac{1}{2^{d_y}}$.

Back to the linear programming (LP) problem (17), it follows from the Fundamental Theorem of Linear Programming (Nocedal & Wright, 1999), that if an LP problem has a unique optimal solution $y^\star \in \mathbb{R}^{d_y}$, the optimal solution must lie at the vertex, which is the intersection of $d$ linearly independent inequality constraints.

By translating the coordinate system with the optimal solution $y^\star$ as the origin, we can find that computing the probability $\mathbb{P}_{y \sim B_\epsilon(y^\star)} (y \in \mathcal{C})$ is actually equivalent to the problem described above. In that case, we have

$$\mathbb{E}_{y \sim B_\epsilon(y^\star)} [\mathbb{P} (y \in \mathcal{C})] = \frac{1}{2^{d_y}}.$$

$\square$

# B. Multiple Sampling Cannot Rescue Supervised Diffusion

It has been analysed that we can improve the solution quality via multiple sampling of diffusion in (Liang & Chen, 2024). However, we need to clarify here that the solution quality cannot be efficiently enhanced with the supervised diffusion training method due to the mismatch between the desired distribution

$$p_d \propto \begin{cases} \exp(-\|y - y^\star\|^2), & y \in \mathcal{C} \\ 0, & otherwise \end{cases}$$

and the actual diffusion target distribution $p_{target} \propto \exp(-\|y - y^\star\|^2)$, as we mentioned in Section 4.

**Lemma 1.** *(Feasibility under Multiple Sampling.) For a supervised diffusion model, even with multiple samples from itself, it is still very hard to obtain a feasible sample, and the probability that all $m$ samples are located outside the constraint region is bounded by*

$$\Pr_{y_1, \cdots, y_m}(\operatorname{argmax} \omega(y; x) < 0 \,|\, y_1, \cdots, y_m \sim \mathcal{D}_\theta) \leq \left(1 - \frac{1}{2^{d_y}} + C_1 e_\theta^{1/4} + C_2 T^{-1/2}\right)^m.$$

*Proof.* According to Theorem 1 in (Liang & Chen, 2024), with $m$ samples from the supervised diffusion model, we have

$$
\begin{aligned}
\Pr(y \sim \mathcal{D}_\theta \in \mathcal{C}) &= \Pr(y \sim \mathcal{D}_\theta \in \mathcal{C}) - \Pr(y \sim \text{dataset} \in \mathcal{C}) + \Pr(y \sim \text{dataset} \in \mathcal{C}) \\
&\geq \Pr(y \sim \text{dataset} \in \mathcal{C}) - |\Pr(y \sim \mathcal{D}_\theta \in \mathcal{C}) - \Pr(y \sim \text{dataset} \in \mathcal{C})| \\
&\geq \Pr(y \sim \text{dataset} \in \mathcal{C}) - \sup_A \left\{ |\Pr(A; \theta) - \Pr(A; \text{dataset})| \right\} \\
&= \Pr(y \sim \text{dataset} \in \mathcal{C}) - \text{TV}(p_\theta(y; x), p_{target}(y; x)) \\
&= \Pr(y \sim \text{dataset} \in \mathcal{C}) - \Pr(y \sim \mathcal{D} \in \mathcal{C}) + \Pr(y \sim \mathcal{D} \in \mathcal{C}) \\
&\qquad\qquad\qquad\qquad - \text{TV}(p_\theta(y; x), p_{target}(y; x)) \\
&\geq \Pr(y \sim \mathcal{D} \in \mathcal{C}) - \text{TV}(p_{target}(y; x), p_d(y; x)) - \text{TV}(p_\theta(y; x), p_{target}(y; x))
\end{aligned}
$$

where $\Pr(y \sim \mathcal{D}_\theta \in \mathcal{C})$ denotes the probability of sampling one feasible solution from supervised diffusion, $\Pr(y \sim \text{dataset} \in \mathcal{C})$ denotes the probability of sampling one feasible solution from the dataset distribution, and $\Pr(y \sim \mathcal{D} \in \mathcal{C})$ denotes the probability of sampling one feasible solution from the desired distribution. $p_\theta(y; x)), p_d(y; x),$ and $p_{target}(y; x)$ represent the probability density of the supervised diffusion model, desired distribution and dataset distribution given condition $x$, respectively. Then, according to (Liang & Chen, 2024), here we can split the total variation distance between the actual diffusion model distribution $p_\theta^{discrete}(y; x)$ and the dataset distribution $p_{taeget}(y; x)$ into three parts:

$$
\text{TV}\left(p_\theta^{discrete}(y; x), p_{target}(y; x)\right) \leq \underbrace{\text{TV}\left(p_{taeget}(y; x); p_\theta(y; x)\right)}_{\text{learning error}} + \underbrace{\text{TV}\left(p_\theta(y; x); p_\theta^{discrete}(y; x)\right)}_{\text{discretization error}}
$$

$$
\leq C_1 e_\theta^{1/4} + C_2 T^{-1/2}
$$

where $C_1, C_2$ are positive constant, $e_\theta$ is the generalization error of noise network and $T$ is the number of diffusion step. The detailed definition can be referred to in Appendix A of (Liang & Chen, 2024).

Besides, applying the Theorem 1, the total variation between $p_{target}$ and $p_d$ can be approximated as

$$
\begin{aligned}
\text{TV}(p_{target}(y; x); p_d(y; x)) &= \int_{y \notin \mathcal{C}} |(p_{target}(y; x) - p_d(y; x)| \, dy \\
&= \int_{y \notin \mathcal{C}} |(p_{target}(y; x)| \, dy \\
&\approx 1 - \frac{1}{2^{d_y}}
\end{aligned}
$$

Finally, we can achieve the probability that there exists no feasible solution under $m$ times sampling from the supervised diffusion

$$
\Pr\left( \underset{y_1, \cdots, y_m \sim \mathcal{D}_\theta}{\text{argmax}} \, \omega(y; x) < 0 \right) \leq \left( 1 - \Pr(y \sim \text{dataset} \in \mathcal{C}) + 1 - \frac{1}{2^{d_y}} + C_1 e_\theta^{1/4} + C_2 T^{-1/2} \right)^m.
$$

Considering all $y$ in the dataset are the optimal solution, we can simplify this formula as

$$
\Pr\left( \underset{y_1, \cdots, y_m \sim \mathcal{D}_\theta}{\text{argmax}} \, \omega(y; x) < 0 \right) \leq \left( 1 - \frac{1}{2^{d_y}} + C_1 e_\theta^{1/4} + C_2 T^{-1/2} \right)^m.
$$

For a high-dimensional problem, $\left( 1 - \frac{1}{2^{d_y}} + C_1 e_\theta^{1/4} + C_2 T^{-1/2} \right)^m$ is obviously very close to 1 even with a sufficiently large number of samples from diffusion. That is why multiple sampling cannot rescue the infeasibility of supervised diffusion.

$\square$

## C. Analysis for Function of Reset Operation

As we mentioned in Section 5, if we continue updating diffusion with the weight of (11), the output of the diffusion solver will resemble that of supervised learning and will be prone to generating infeasible points. For simplicity, we consider the

case that the diffusion solver has been well trained and all the points generated by it are feasible in one optimization problem parameterized by $\boldsymbol{x}$. Then, the weight function will actually be

$$\omega(\boldsymbol{y}) = \exp\left(f^{\star}(\boldsymbol{x}) - f(\boldsymbol{y}; \boldsymbol{x})\right). \tag{18}$$

Let the initial distribution of generated points from the well-trained diffusion solver be $\rho_0(\boldsymbol{y}; \boldsymbol{x})$. Then, after $N$ iterations of weighted bootstrapping using (18), the distribution of generated points will be

$$
\begin{aligned}
\rho_N(\boldsymbol{y}; \boldsymbol{x}) &\propto \rho_0(\boldsymbol{y}; \boldsymbol{x}) \prod_{i=1}^{N} \exp\left(\beta\left(f^{\star}(\boldsymbol{x}) - f(\boldsymbol{y}; \boldsymbol{x})\right)\right) \\
&= \rho_0(\boldsymbol{y}; \boldsymbol{x}) \exp\left(\beta N \cdot \left(f^{\star}(\boldsymbol{x}) - f(\boldsymbol{y}; \boldsymbol{x})\right)\right),
\end{aligned}
\tag{19}
$$

according to the reweighting technique, where $\beta > 0$ is a small value determined by the learning rate. Hence, when $N \to \infty$, $\rho_N(\boldsymbol{y}; \boldsymbol{x})$ will converge to the Dirac distribution $\delta(x - x^{\star})$. This is equivalent to the supervised diffusion solver trained with optimal points. In contrast, we can avoid this problem by redistributing the probability density across the feasible region with $\rho_1 \propto \rho_0 \exp(\beta\left(f^{\star}(\boldsymbol{x}) - f(\boldsymbol{y}; \boldsymbol{x})\right)) \approx \rho_0$ using the reset operation.

In addition, we provide an experiment in Table 5 to illustrate the impact of Reset. As shown in the table below, not applying Reset leads to varying degrees of feasibility degradation in DiOpt. The underlying reason is exactly as explained in (19).

| Problem | Reset | Feasibility(%)↑ |
|---------|-------|-----------------|
| QPSR | False | $70.03\% \pm 0.00$ |
|      | True | $\mathbf{81.19\% \pm 0.00}$ |
| CQP | False | $83.79\% \pm 0.00$ |
|     | True | $\mathbf{87.07\% \pm 0.00}$ |

Table 5. Effects of Reset on QPSR and CQP Tasks. On all three tasks, applying Reset achieves better feasibility.

## D. Hyperparameter Settings

In this paper, the following hyperparameters are involved:

| Parameter | Symbol | Value (Main Experiments) |
|-----------|--------|--------------------------|
| Supervised learning ratio | $r_s$ | 0.2 |
| Number of evaluation points | $K_e$ | 64 |
| Number of training samples | $K_t$ | 16 |
| Number of training epochs | $N_e$ | 10000 |
| Number of diffusion steps | $T$ | 100 |
| Noise schedule coefficient | $\eta$ | 1 |

Table 6. Experimental hyperparameters used in the main text.

The number of sampling points refers to the number of samples generated by the model $\mathcal{D}_\theta$ on the validation and test sets. The number of training samples, $K_t$, denotes the number of samples used during training. Note that $K_t \neq K_e$, as a large training sample size would significantly increase training time. The supervised learning ratio $r_s$ determines the portion of training steps that use supervised learning. Specifically, DiOpt is trained with optimal solutions during the first $\lfloor N_e \cdot r_s \rfloor$ steps. After that, the buffer $\mathcal{B}$ is initialized, and training continues in the same manner. By setting $r_s = 1$, the procedure reduces to standard Diffusion as described in the paper. The number of diffusion steps corresponds to the variable $T$ in (6).

Our noise network $\epsilon_\theta$ consists of a time encoding module and a backbone network. The time encoding module maps each timestep into a 32-dimensional vector using sinusoidal positional embeddings, followed by a two-layer MLP with 512 hidden units and Mish activation. The backbone network comprises four linear layers, each with 512 hidden units. The input dimension is $d_y + d_x + 32$. Each layer is followed by a Mish activation function. The final output of diffusion model has dimension $d_y$. In all experiments, we apply equality constraint completion for all baselines, but do not perform inequality correction for diffusion-based methods.

The experiments were performed on a high-performance workstation featuring an Intel Core i9-14900K processor (24 cores, 32 threads, 6.0 GHz turbo frequency) paired with dual NVIDIA GeForce RTX 4090D GPUs (24GB GDDR6X VRAM each) for accelerated deep learning computations. The system was equipped with 128GB DDR5 RAM. The Ubuntu 22.04.5 LTS operating system with Linux kernel 6..8.0-79-generic and NVIDIA driver 550.163.01 provided an optimized environment for GPU-accelerated workloads.

# E. Results on Convex Problem

*Table 7.* Results on QP benchmark. (mean $\pm$ std. dev.) across different methods. The row highlighted in light blue indicates a "solver" (IPOPT here).

| Problem | Method | Feasibility(%)↑ | Gap(%)↓ | Objective↓ | Time(s)↓ | Ineq Mean↓ | Ineq Max↓ | Ineq Num Viol↓ |
|---------|--------|-----------------|---------|------------|----------|------------|-----------|----------------|
| | IPOPT | $100.00\% \pm 0.00$ | $0.00\% \pm 0.00$ | $-10.90 \pm 0.45$✓ | $0.63 \pm 0.11$ | $0.00 \pm 0.00$ | $0.00 \pm 0.00$ | $0.00 \pm 0.00$ |
| | RectFlow | $0.00\% \pm 0.00$ | $0.97\% \pm 3.22$ | $-10.81 \pm 0.55$✗ | $0.01 \pm 0.00$ | $0.01 \pm 0.03$ | $0.58 \pm 0.65$ | $14.02 \pm 4.20$ |
| QP | DC3 | $22.93\% \pm 0.00$ | $34.98\% \pm 6.64$ | $-7.10 \pm 0.83$✗ | $0.01 \pm 0.00$ | $0.00 \pm 0.01$ | $0.48 \pm 0.64$ | $2.09 \pm 2.34$ |
| | MBD | $0.04\% \pm 0.00$ | $3588.43\% \pm 3743.72$ | $380.26 \pm 409.20$✗ | $0.01 \pm 0.00$ | $10.77 \pm 7.47$ | $81.30 \pm 50.46$ | $95.14 \pm 19.30$ |
| | DiOpt* | $\mathbf{79.11\% \pm 0.00}$ | $\mathbf{3.45\% \pm 1.23}$ | $-10.53 \pm 0.47$✓ | $0.01 \pm 0.00$ | $0.00 \pm 0.00$ | $\mathbf{0.02 \pm 0.06}$ | $\mathbf{0.28 \pm 0.50}$ |

As an additional validation, we evaluate DiOpt on a convex quadratic programming (QP) problem,with the results reported in Table 7. We observe that DiOpt again achieves the highest feasibility among all baselines. In contrast, RectFlow attains the smallest optimality gap but exhibits a notably low feasibility rate. This behavior is consistent with the theoretical analysis presented in Lemma 1. Details for QP benchmark can be founded in Appendix G.

# F. Ablation Study

In this section, we conduct a series of ablation experiments to analyze the impact of various hyperparameters on the experimental results. Appendix F.1 investigates the effect of $T$. Appendix F.2 examines the impact of $r_s$. Appendix F.3 studies the influence of $K_t$. Appendix F.4 reports the performance under different $K_e$. Appendix F.5 presents the training dynamics of various models. Appendix F.6 evaluates the effect of $\eta$. Appendix F.7 investigates the necessity of the look-up table by comparing performance with and without it on the QPSR benchmark. Appendix F.8 validates our proposed weighting mechanism against a naive penalty-based weighting baseline, demonstrating the superiority of our design. Unless otherwise specified, the default parameters in this section are set as: $N_e = 10000, r_s = 20\%, K_e = 32, K_t = 16, \eta = 0$.

## F.1. Different Diffusion Steps

Recent studies (Song et al., 2020a) have pointed out that the number of diffusion steps can significantly affect the quality of the generated solutions. In addition, according to (6), different values of $T$ introduce different levels of noise into the sampling process, which also affects the discretization error in Lemma 1.

To investigate the effect of diffusion steps $T$ on performance, we visualize the training dynamics of DiOpt under different diffusion steps in Figure 5. As shown in the figure, for the four tasks considered, the performances of $T = 10$ and $T = 20$ vary across tasks. Only $T = 100$ consistently achieves the smallest optimality gap across all tasks. Furthermore, for all values of $T$, a significant improvement in feasibility is observed immediately after the bootstrapping phase begins (around 2000 epochs). This observation is consistent with the conclusions drawn in the main text.

## F.2. Different Supervised Ratios

Here $r_s$ critically determines how closely DiOpt approaches the optimal value before bootstrapping initialization. It should be noted that due to the curse of dimensionality (Bellman et al., 1957), the sampling capability of Diffusion-based methods under limited computational resources cannot guarantee sufficient exploration of the solution space. This limitation often leads to suboptimal objective function performance when supervision is inadequate.

As demonstrated in Figure 6, models with $r_s = 0.05, 0.02$, and $0.2$ achieve comparable feasibility performance. However, across both QP and QPSR tasks(with Dimension 100), the $r_s = 0.02$ configuration demonstrates relatively inferior objective function performance compared to $r_s = 0.05, 0.2$, attributable to insufficient supervised learning. In contrast, both $r_s = 0.05, 0.2$ maintain objective function values comparable to Diffusion while achieving satisfactory feasibility.

*(a)* ACOPF57

**ACOPF57**

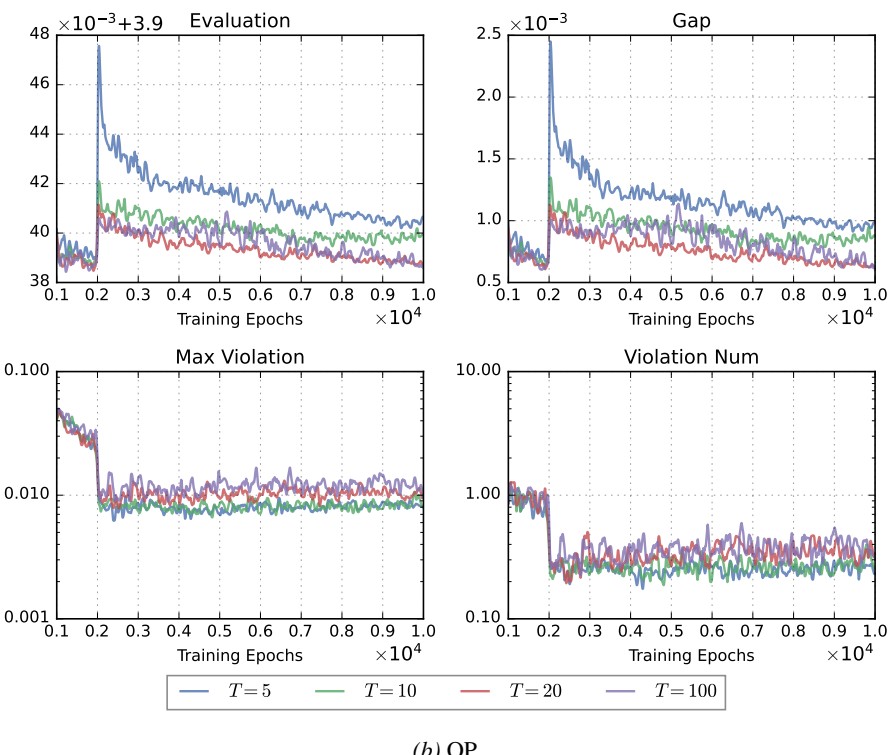

*(b)* QP

**QP**

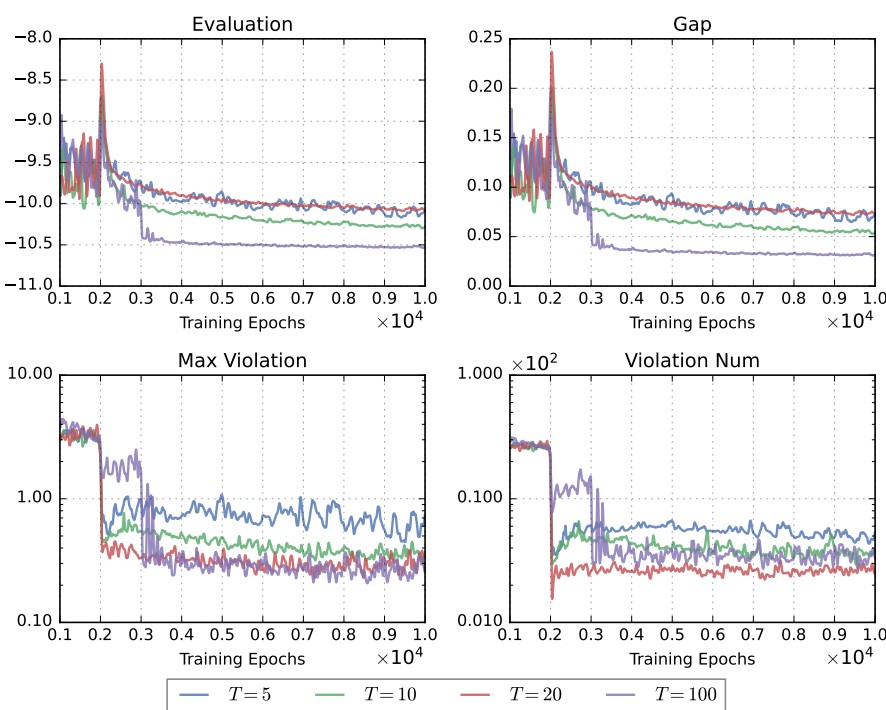

*(c)* QPSR

**QPSR**

*(d)* CQP

**CQP**

*Figure 5.* **Effects of Different Diffuion Steps**. In this experiment, we examine how varying the number of diffusion steps $T$ affects performance. All other hyperparameters are fixed as follows: $r_s = 0.2$, $K_t = 16$, and $K_e = 32$ for all values of $T$.

These results highlight the importance of selecting an appropriate supervised ratio for DiOpt. In this work, we set the default $r_s$ to 0.2 to ensure: (1) model convergence before bootstrapping, and (2) a clear demonstration of effectiveness of bootstrapping through performance improvements.

### F.3. Different Training Samples

During the training phase of DiOpt, we set $K_t = 1$ for the supervised learning stage. Once bootstrapping begins, $K_t$ is set to 16. Figure 4 demonstrates that DiOpt maintains performance parity with Diffusion prior to bootstrapping, confirming that no additional $K_t$ adjustment is required during supervised learning.

Figure 7 reveals that increased $K_t$ values yield improved objective function performance at the cost of slight feasibility degradation. This trade-off emerges because larger $K_t$ values concentrate the sampling distribution of model nearer to optimal points, consequently narrowing the feasible region. However, given the negligible impact on overall feasibility, we establish $K_t = 16$ as the default configuration of DiOpt. The decision against higher $K_t$ values stems from computational overhead, as they would proportionally increase both training duration and GPU memory requirements.

*(a)* QPSR

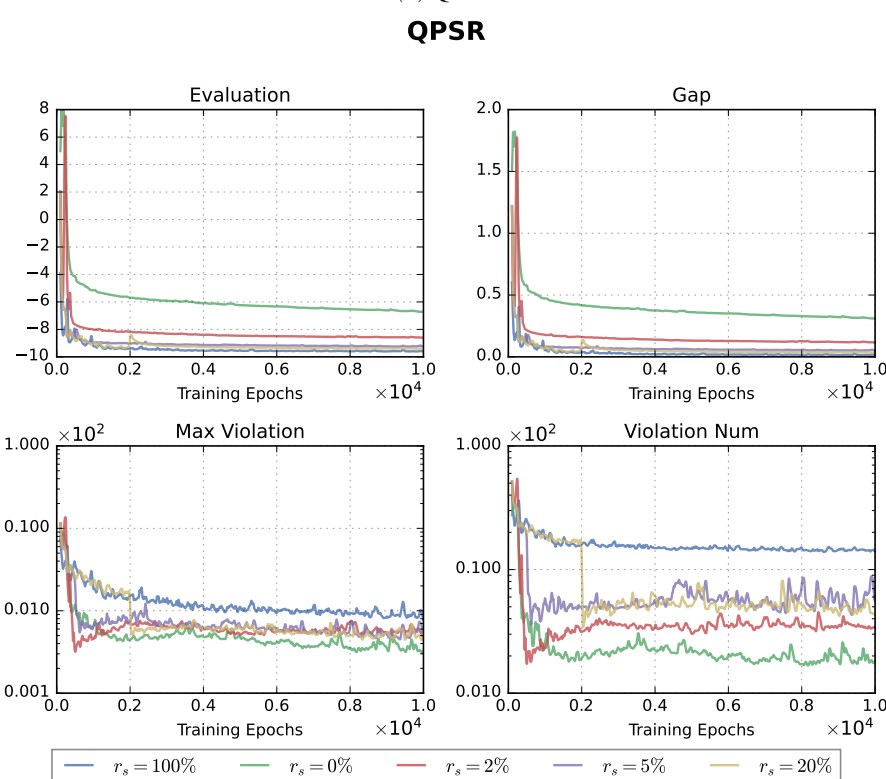

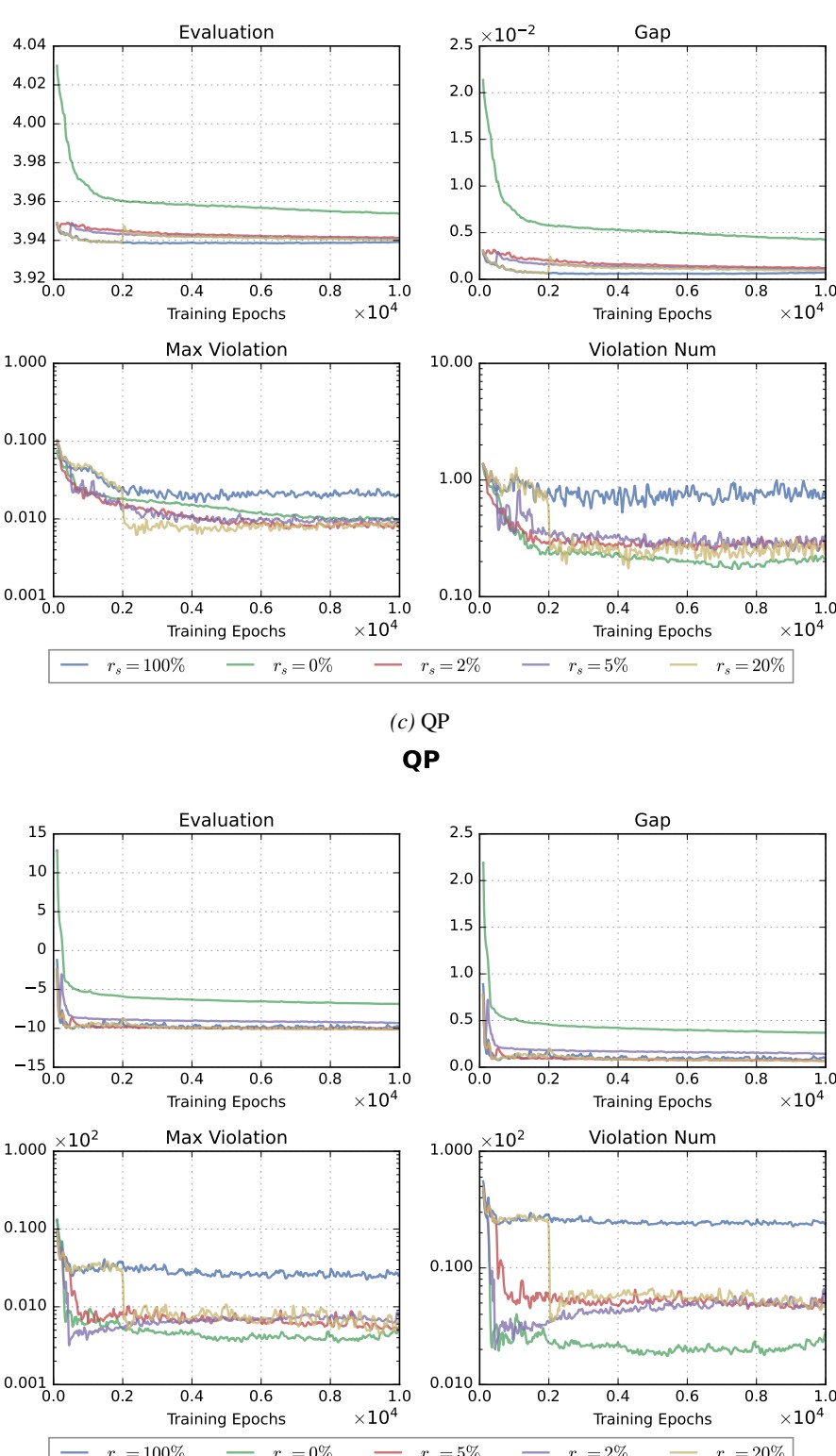

*Figure 6.* **Effects of Different Supervised Ratio**. In this experiment, we examine how varying the number of supervised ratio $r_s$ affects performance. All other hyperparameters are fixed as follows: $T = 5$, $K_t = 16$, and $K_e = 32$ for all values of $r_s$.

*(a)* QPSR

**QPSR**

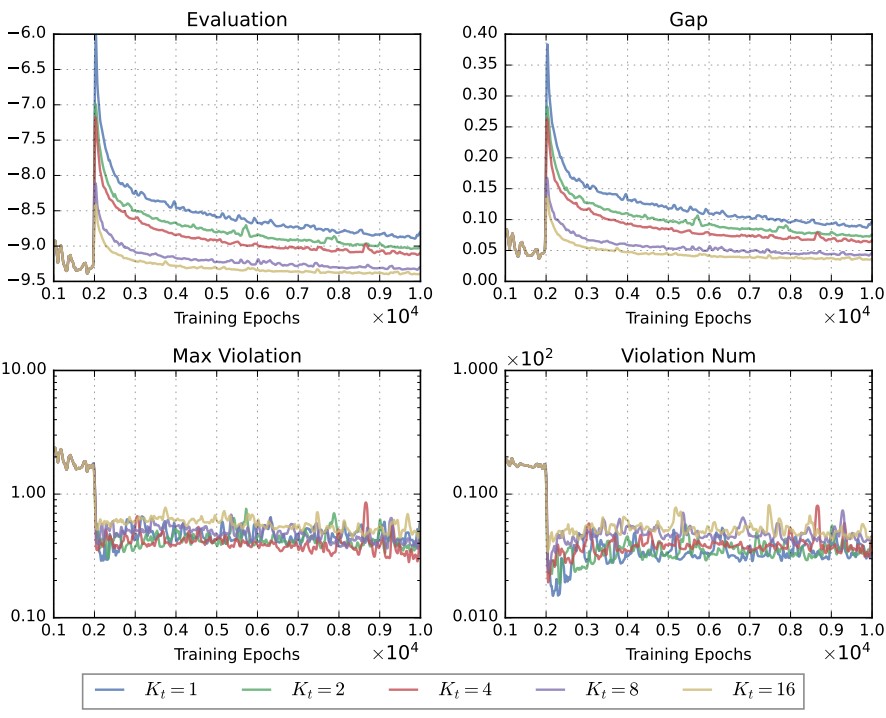

*(b)* ACOPF57

**ACOPF57**

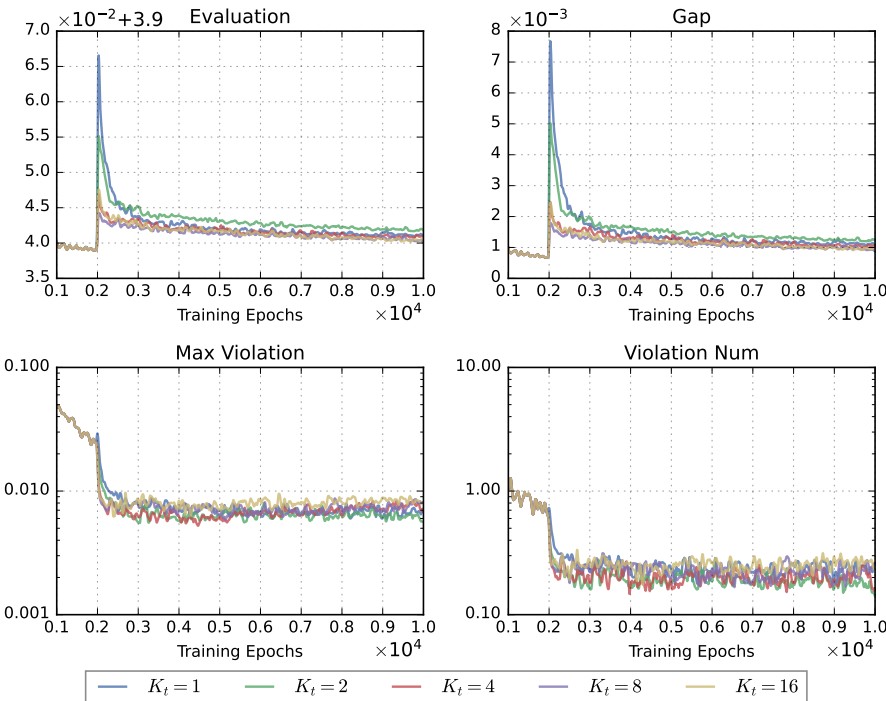

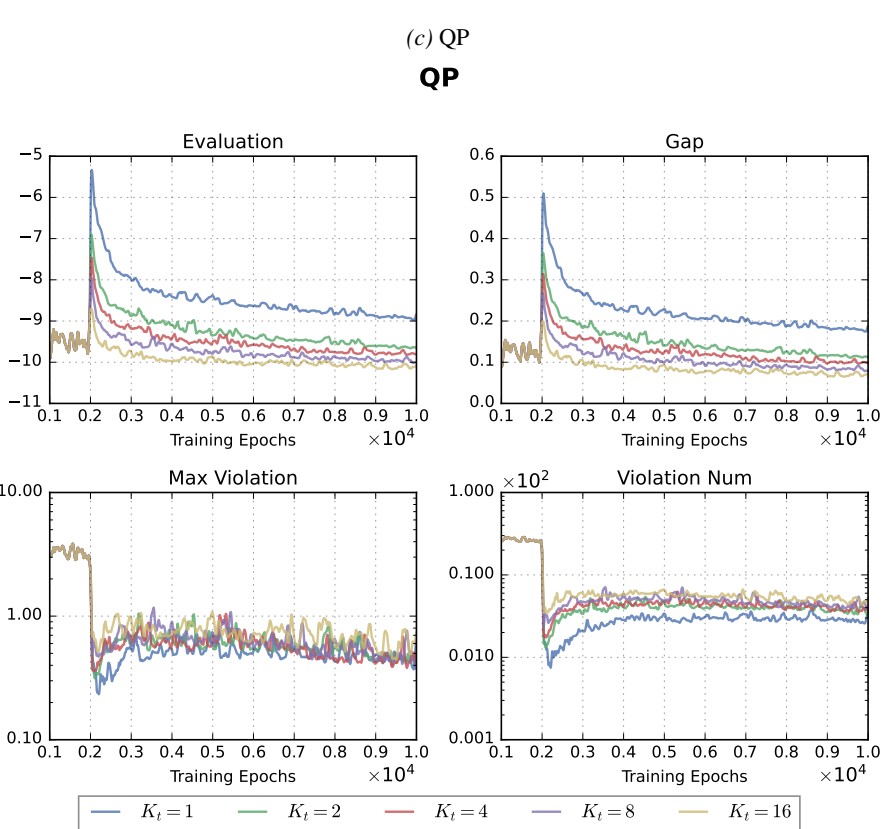

*(c)* QP

*Figure 7.* **Effects of different Training Samples**. In this experiment, we examine how varying the number of training samples $K_t$ affects performance. All other hyperparameters are fixed as follows: $r_s = 0.2$, $T = 5$, and $K_e = 32$ for all values of $K_t$.

## F.4. Different Evaluation Points

Here, we report the performance of Diffusion and DiOpt under different values of $K_e$. It can be observed that, in Figure 8b, Diffusion fails to obtain feasible solutions even when increasing $K_e$. In contrast, as observed in Figure 8a, DiOpt is able to generate feasible solutions while maintaining a small optimality gap in tasks with higher constraint dimensions. This demonstrates that the bootstrapping-based training enables the diffusion model to learn the location of the feasible region.

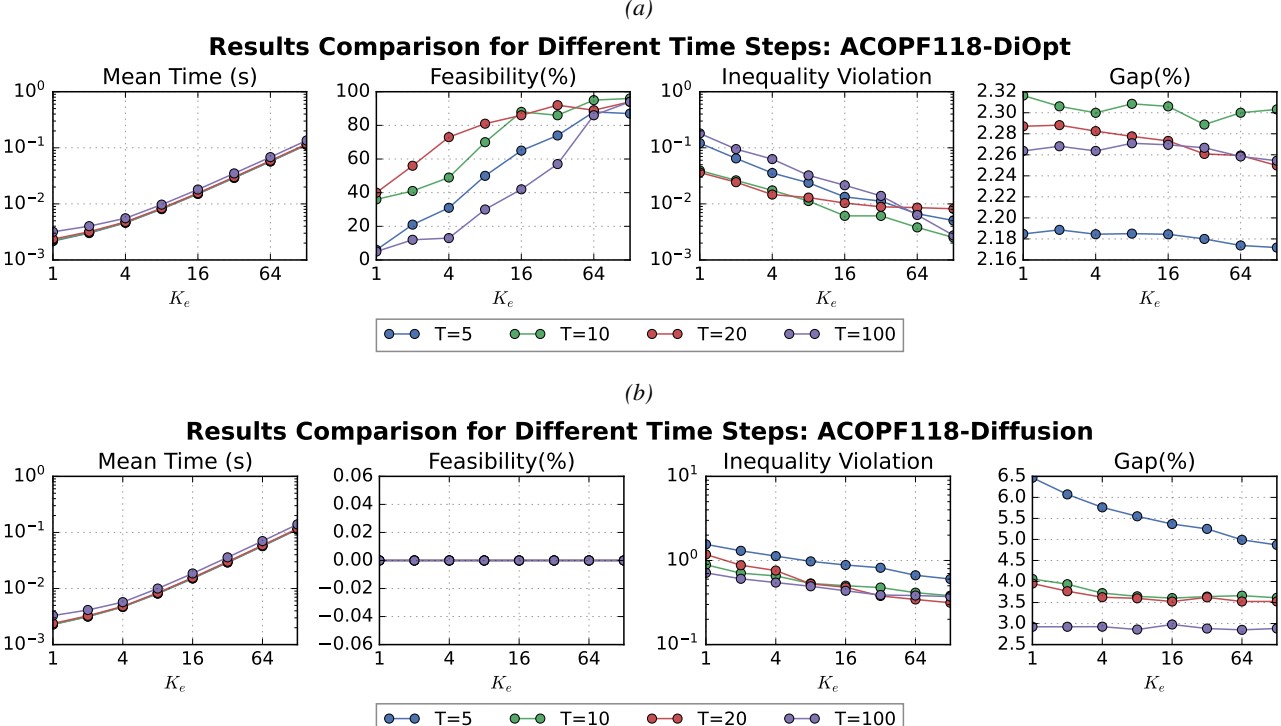

*Figure 8.* **Effects of number of sampling samples**. In this experiment, we examine how varying the number of sampling points $K_e$ affects performance with $\eta = 1$.

## F.5. Different Training Procedures

In this subsection, we primarily analyze the training dynamics of DC3, RectFlow, and DiOpt in the main text. For all experiments in this section, hyperparameters are uniformly set as $K_e = 32$, $K_t = 16$, $r_s = 0.2$, and $T = 5$. Notably, the noise level $\eta$ is configured as 0 in all figures presented, a setting that significantly impacts performance outcomes. This aspect will be further elaborated in Section F.6.

From the Figure 9a and 9b analyses, we observe that both DiOpt and RectFlow exhibit superior performance gaps compared to DC3 throughout the training process, while maintaining comparable constraint violation levels. For the CQP problem (Detailed in Appendix G.5), DC3 exhibits particularly poor performance due to the inherent characteristics of its objective function, where larger $\boldsymbol{y}$ values correspond to steeper gradients. This gradient amplification phenomenon causes DC3 to deviate progressively from feasible regions during iterative updates, ultimately resulting in severe feasibility violations and substantial optimality gaps. In contrast, both RectFlow and DiOpt demonstrate remarkable resilience to such gradient interference, suggesting that diffusion-based optimization frameworks are inherently more robust against sampling point divergence caused by high-gradient objective functions.

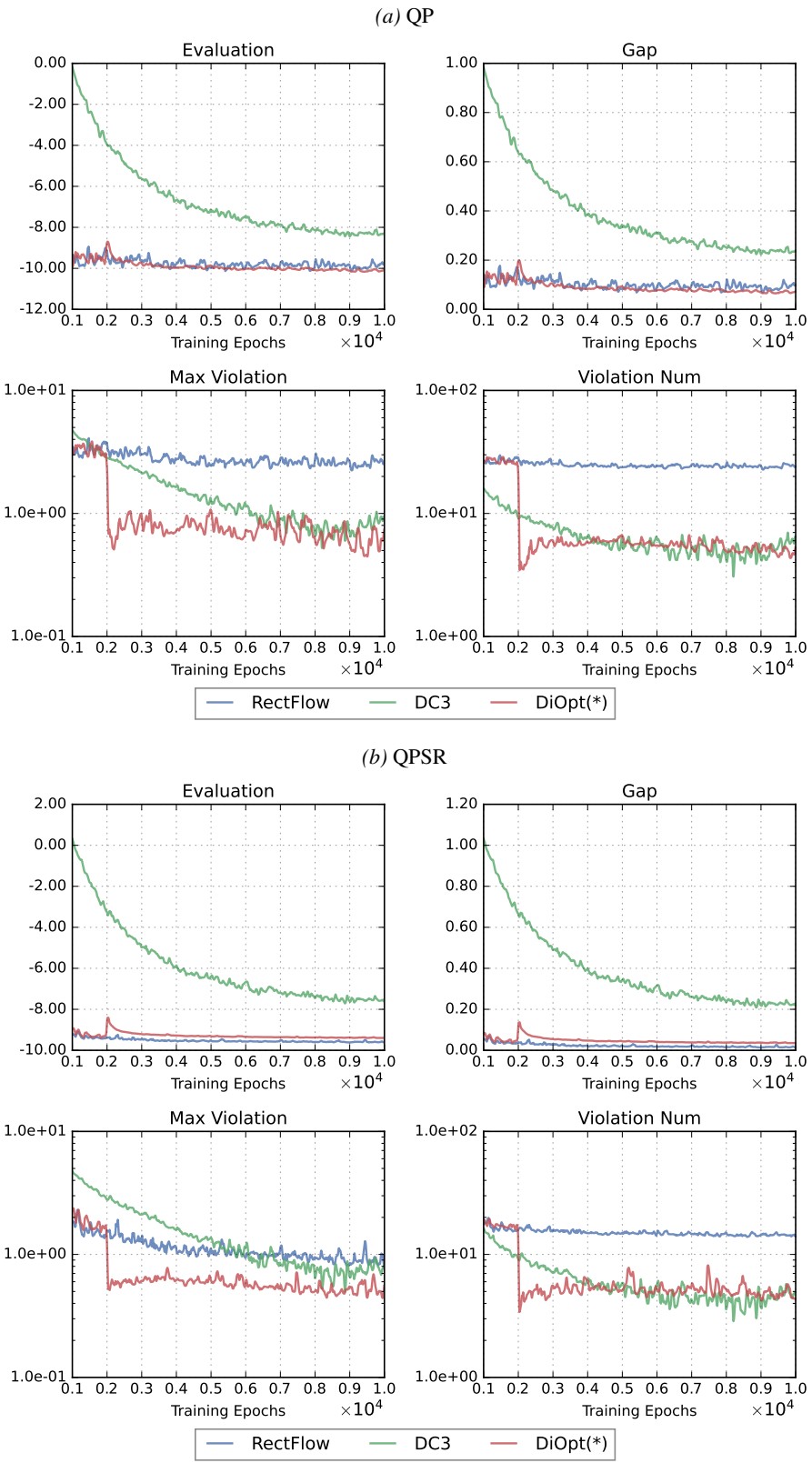

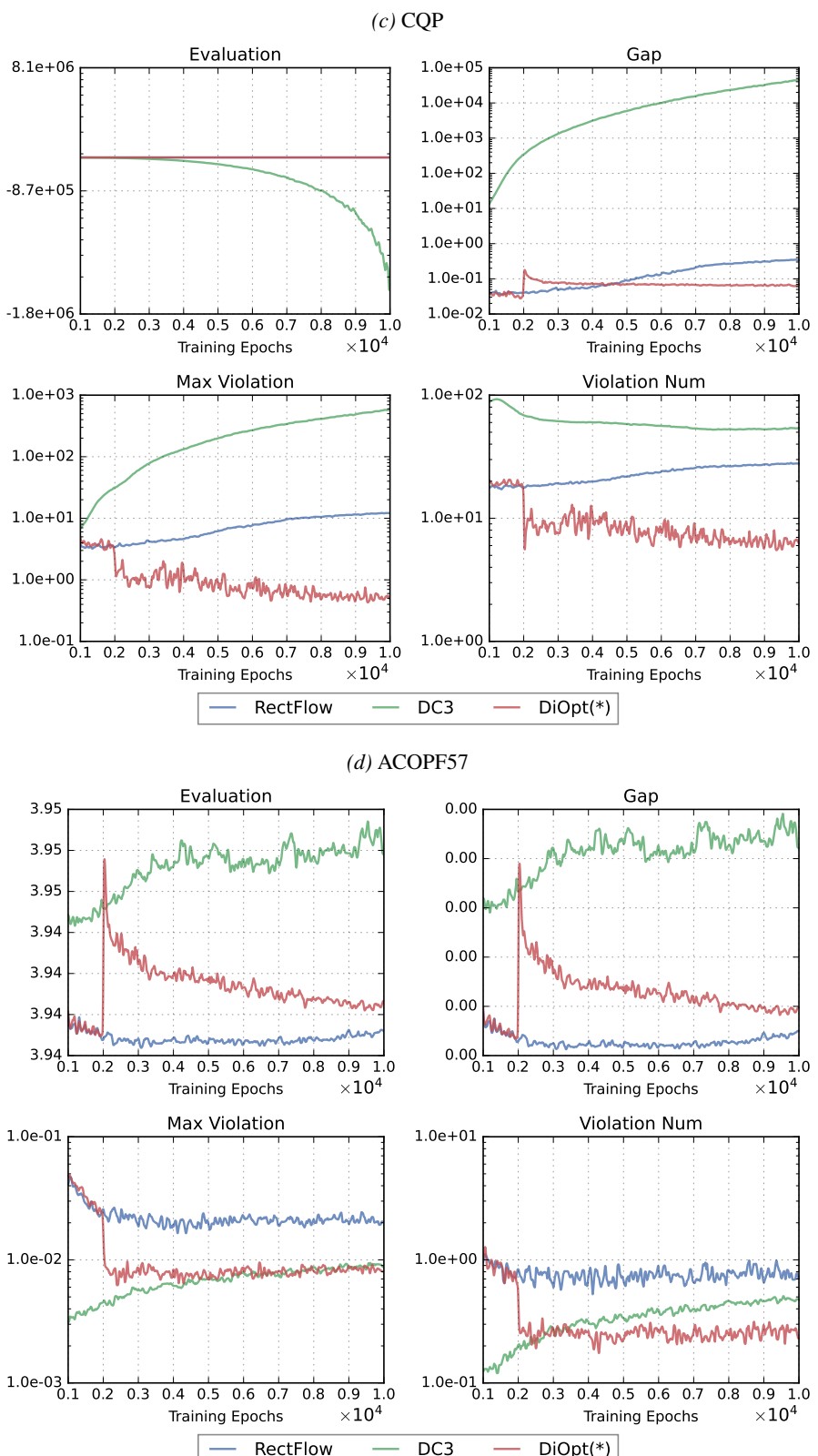

*Figure 9.* **Training Procedure of DiOpt, RectFlow and DC3.** In this experiment, all hyperparameters are fixed as follows: $r_s = 0.2$, $T = 5$, $K_t = 16$, and $K_e = 32$.

## F.6. Different Noise Levels

In this section. we investigate the influence of noise level $\eta$ on feasibility rates. As previously discussed, varying $\eta$ determines the "exploration range" of the diffusion model, which significantly impacts its performance on optimization tasks. As shown in Figure 10 and Figure 11, setting $\eta = 0$ implies that even varying $K_e$ does not affect the results. It can be observed that the effect of $\eta$ is not entirely consistent across different values of $K_e$. As shown in the figure, there exists an optimal interval of $\eta$ within which feasibility is maximized. When $\eta$ lies close to this optimal interval, the feasibility performance is the best; conversely, when $\eta$ deviates from this interval, the feasibility deteriorates. The location of this optimal interval varies with $K_e$. For instance, when $K_e = 256$, the optimal interval is approximately $1.5 < \eta < 2$. In contrast, for $K_e = 1$, the optimal threshold is attained at $\eta = 0$. Considering resource consumption, setting $K_e = 64$ with $\eta \in [0.9, 1.2]$ is empirically sufficient.

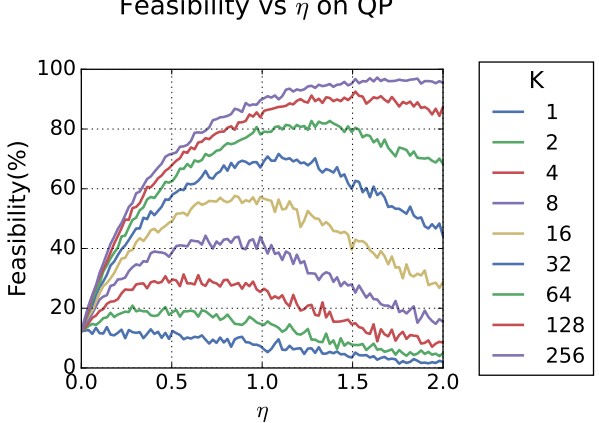

*Figure 10.* Feasibility of DiOpt under different $\eta$ and $K_e$.

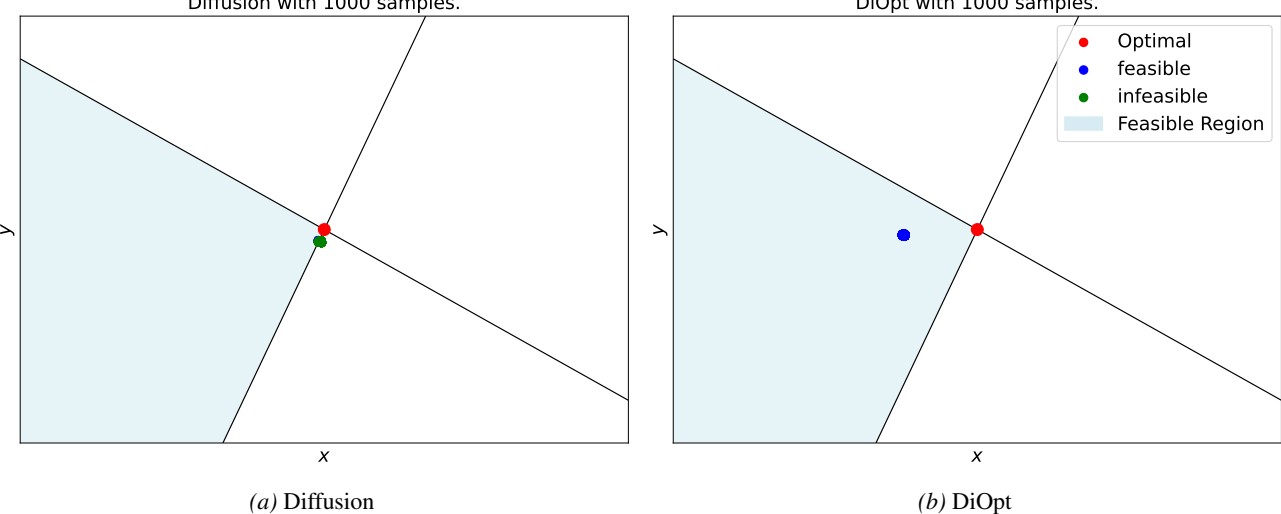

*(a)* Diffusion  *(b)* DiOpt

*Figure 11.* For the $\eta = 0$ configuration in the toy example, observations reveal that despite initial sampling starting from Gaussian-distributed random noise, all trajectories converge to nearly identical positions after denoising when $\eta = 0$. This demonstrates the necessity of noise injection during the sampling process.

## F.7. Effect of the Look-up Table

To demonstrate the necessity of the look-up table in the second-stage refinement, we conduct an ablation study on the QPSR benchmark. Without the look-up table, the model directly suffers from poor solutions caused by the inherent stochasticity of the sampling process, which cannot guarantee that each sample is better than the previous one. The results are reported in Table 8. As shown in Table 8, removing the look-up table leads to a dramatic degradation in both optimality gap and

*Table 8.* Ablation study on the look-up table for QPSR. "Gap" measures the relative optimality gap and "Violation" measures the average constraint violation.

| Configuration | Gap ↓ | Violation ↓ |
|---|---|---|
| w. look-up table | **6.69**% | **0.008** |
| w.o. look-up table | 112.8% | 0.120 |

constraint violation, confirming that the look-up table is essential for ensuring consistent improvement during bootstrapping.

### F.8. Comparison with Penalty-Based Weighting

To further validate the effectiveness of our proposed weighting mechanism, we compare it against a naive penalty-based weighting baseline. The penalty-based weight is defined as:

$$w_{\text{penalty}}(y; x) = \exp\left(-f(y; x) - 10 \cdot \sum_i \max(g_i(y; x), 0)\right),\tag{20}$$

where $f(y; x)$ is the objective function and $g_i(y; x) \leq 0$ are the inequality constraints. This baseline assigns an exponentially decaying weight when constraints are violated. The empirical comparison on QPSR is presented in Table 9. The results

*Table 9.* Comparison of our weighting mechanism with the penalty-based baseline on QPSR.

| Weight Type | Gap ↓ | Violation ↓ |
|---|---|---|
| Ours | **6.69%** | **0.008** |
| Penalty (additional baseline) | 8.17% | 0.03 |

show that the penalty-based weighting leads to larger optimality gaps and higher constraint violations compared to our mechanism. This demonstrates that our weighting scheme, which carefully balances both objective improvement and feasibility, is superior to a naive penalty-based approach.

## G. Benchmark configuration

In this section, we provide detailed descriptions of some tasks referenced in the main text. These include concave quadratic optimization problems (CQP), relatively complex nonconvex optimization problems with practical significance, and others.

### G.1. Benchmark Details

We present a table summarizing the relevant parameters for each benchmark used in Section 6:

*Table 10.* Benchmark parameters used in our experiments.

| **Problem** | $d_x$ | $d_y$ | $d_z$ | $m$ | $n$ | training set | validation set | test set | active constraints |
|---|---|---|---|---|---|---|---|---|---|
| QP | 50 | 100 | 50 | 250 | 50 | 8334 | 833 | 833 | 30.30(2.03) |
| QPSR | 50 | 100 | 50 | 250 | 50 | 8334 | 833 | 833 | 24.18(2.01) |
| CQP | 50 | 100 | 50 | 250 | 50 | 8334 | 833 | 833 | 51.55(1.18) |
| ACOPF57 | 114 | 128 | 13 | 142 | 114 | 1000 | 100 | 100 | 6.01(1.76) |
| ACOPF118 | 236 | 344 | 107 | 452 | 236 | 1000 | 100 | 100 | 22.70(2.49) |
| Retargeting | 39 | 19 | 19 | 39 | 0 | 1447 | 145 | 145 | 0.97(0.18) |

According to DC3 (Donti et al., 2021), the neural network produces an output vector $z \in \mathbb{R}^{d_z}$. This output is then completed via equality constraints to form the optimization variable

$$y = [\phi_x^\top(z), z^\top]^\top \in \mathbb{R}^{d_y},$$

which satisfies the equality constraints $h(y; x) = 0$. The column 'Active Constraints' reports the mean number of inequality constraints that are active ($\exists i$ s.t. $g_i(y; x) = 0$) in our dataset for each benchmark. These inequality constraints must be directly satisfied by the neural network, with certain constraints enforced by interpolating between their minimum and maximum values (like box constraints in Retargeting).

### G.2. Toy Example

The Toy Example shown in Figure 2 is defined as follows:

$$\min_{y_1, y_2} \quad f(y_1, y_2) = \left(y_1 - \frac{65}{19}\right)^2 + \left(y_2 - \frac{24}{19}\right)^2$$
$$\text{s.t.} \quad -4y_1 - 3y_2 \leq -12,$$
$$-y_2 \leq 0,$$
$$4y_1 + 5y \leq 20,$$
$$0 \leq y_1, y_2 \leq 5. \tag{21}$$

The Optimal Point (point marked as red) in Figure 2 is $\left(\frac{65}{19}, \frac{24}{19}\right)$.

### G.3. Quadratic Programming (QP)

The Simple Problem discussed in the text is defined as follows:

$$\min_{y \in \mathbb{R}^n} \quad \frac{1}{2} y^T Q y + p^T y,$$
$$\text{s.t.} \quad Ay = x,$$
$$Gy \leq h. \tag{22}$$

Here, $x$ is treated as a conditional parameter of the optimization problem, sampled uniformly from $[-1, 1]$ across all instances. $Q$, $p$, $A$, and $h$ remain fixed. $Q$ is a diagonal matrix whose diagonal elements are independently and identically sampled from $[0, 1]$. The vector $p$ is generated using the same method as $Q$. Elements of matrices $A$ and $G$ are sampled from a standard normal distribution. To ensure the feasibility of $Gy \leq h$, $h$ is constructed as:

$$h_i = \sum_j |(GA^+)_{ij}|, \tag{23}$$

where $A^+$ denotes the pseudoinverse of $A$. The optimal solutions are generated through IPOPT (Wächter & Biegler, 2006). **10000 examples have been generated for this task.**

### G.4. Quadratic Programming with Sine Regularization (QPSR)

The Nonconvex Problem in the text is formulated as:

$$\min_{y \in \mathbb{R}^n} \quad \frac{1}{2} y^T Q y + \alpha \cdot p^T \sin(y),$$
$$\text{s.t.} \quad Ay = x,$$
$$Gy \leq h. \tag{24}$$

Here, $x$ similarly serves as a conditional parameter, and the generation methods for $x$, $Q$, $p$, $A$, $G$, and $h$ align with those in the Simple Problem. In our experiment, $\alpha$ was setted as 1. The optimal solutions are generated through IPOPT. **10000 examples have been generated for this task.**

### G.5. Concave Quadratic Programming (CQP)

The CQP discussed in the text is defined as follows:

$$\min_{y \in \mathbb{R}^n} \quad \frac{1}{2} y^T Q y + p^T y,$$
$$\text{s.t.} \quad Ay = x,$$
$$Gy \leq h. \tag{25}$$

Here, $x$ is treated as a conditional parameter of the optimization problem, sampled uniformly from $[-1, 1]$ across all instances. $Q$, $p$, $A$, and $h$ remain fixed. $Q$ is a diagonal matrix whose diagonal elements are independently and identically sampled from $[-1, 0]$. The vector $p$ is generated using the same method as $Q$. Elements of matrices $A$ and $G$ are sampled from a standard normal distribution. To ensure the feasibility of $Gy \leq h$, $h$ is constructed as:

$$h_i = \sum_j |(GA^+)_{ij}|, \tag{26}$$

where $A^+$ denotes the pseudoinverse of $A$. The optimal solutions are generated through IPOPT. **10000 examples have been generated for this task.** It is important to note that this task is deliberately designed to challenge gradient-based methods. Specifically, it exhibits the property

$$\lim_{\|y\| \to \infty} \|\nabla f(y; x)\| = \infty,$$

which implies that the gradient norm diverges as $\|y\|$ grows. Consequently, methods such as DC3, which rely on $\nabla f(y; x)$ to train the neural network, fail on this problem.

### G.6. ACOPF

The AC Optimal Power Flow (ACOPF) (Cain et al., 2012; Shi et al., 2017) is a core problem in power systems, aiming to minimize generation costs by adjusting active/reactive power outputs of generators, voltage magnitudes, and phase angles while satisfying constraints such as power balance, line flow limits, and voltage limits. Although the generation costs are merely simple quadratic functions, the intricate constraints render the ACOPF a highly non-convex problem. This results in traditional solution algorithms for ACOPF encountering issues such as global optimality and excessive computation times etc. Recent studies have proposed relaxation approaches (Bingane et al., 2018) and machine learning-based approaches (Zamzam & Baker, 2020; Zhang & Zhang, 2022; Jiang et al., 2024; Zhao & Barati, 2024) to address these issues.

More specifically, an ACOPF problem involves $N$ nodes, including load buses $\mathcal{L}$, a reference bus $\mathcal{R}$, and generator buses $\mathcal{G}$. Variables include active power $p_g$, reactive power $q_g$, active demand $p_d$, reactive demand $q_d$, voltage magnitude $|v|$, and voltage phase angle $\theta$. Load buses (representing non-generating nodes) satisfy $(p_g)_\mathcal{L} = (q_g)_\mathcal{L} = 0$. The reference bus provides a phase angle reference, with $\theta_\mathcal{R} = \theta_{\text{ref}}$. Network parameters are described by the admittance matrix $Y$. The ACOPF is formalized as follows, where $v = |v|e^{i\theta}$, and $A$, $b$ are fixed parameters related to generation costs:

$$
\begin{aligned}
\min_{p_g, q_g, v, \theta} \quad & p_g^T A p_g + b^T p_g, \\
\text{s.t.} \quad & \underline{p}_g \leq p_g \leq \overline{p}_g, \\
& \underline{q}_g \leq q_g \leq \overline{q}_g, \\
& \underline{|v|} \leq |v| \leq \overline{|v|}, \\
& \theta_\mathcal{R} = \theta_{\text{ref}}, \\
& (p_g)_\mathcal{L} = (q_g)_\mathcal{L} = 0, \\
& (p_g - p_d) + i(q_g - q_d) = \text{diag}(v) Y v^*.
\end{aligned}
\tag{27}
$$

where $A, b$ represent as the cost coefficient, the underline represent the lower bound, and overline represent the upper bound. In this formulation, nodal demands $p_d$ and $q_d$ act as conditional parameters. Our experiments test the 57-bus system (ACOPF57) and 118-bus system (ACOPF118), with optimal solutions obtained via IPOPT (Wächter & Biegler, 2006). **1200 problems are generated for both ACOPF57 and ACOPF118**.

### G.7. Retargeting Problem

The motion retargeting task can be formulated as an optimization problem, where the objective is to minimize the discrepancy between the motion of the SMPL human model and the H1 robot model. This task involves not only the alignment of joint positions but also the consideration of differences in kinematic structure, body proportions, joint alignment, and end-effector

positioning. Due to the significant differences between the kinematic structure of the SMPL model and the kinematic tree of the H1 humanoid robot, He et al. (2024) proposed a two-step method for preliminary motion retargeting. In the first step, given that the body shape parameters $\beta$ of the SMPL human model can represent a variety of body proportions, we optimize to find a body shape $\beta'$ that best matches the humanoid robot's structure, thereby minimizing the joint position discrepancies between the models. This ensures that the joint positions of the SMPL model and H1 robot align as closely as possible, laying the foundation for subsequent retargeting.

Once the optimal $\beta'$ is determined, the second step involves mapping the joint positions and postures of the H1 robot to their corresponding positions in the SMPL model using forward kinematics. This process takes into account the kinematic constraints of the robot, ensuring the validity of joint positions. Finally, to further refine the joint alignment, we minimize the differences in the positions of 11 key joints, adjusting the joint configuration between the SMPL model and the H1 robot. It is important to note that the retargeting process goes beyond adjusting joint positions—it also involves the alignment of end-effectors (such as ankles, elbows, and wrists). Special attention is given to the precise alignment of these key points to ensure that the human motion is smoothly transferred to the humanoid robot. Given a sequence of motions expressed in SMPL parameters, which takes as input the joint positions $\boldsymbol{P}_{SMPL}$, root rotation $\boldsymbol{R}_{root}$, and transform offset $\boldsymbol{O}_{offset}$ from the SMPL model, and computes the global joint positions $\boldsymbol{P}_{H1}$ of the H1 robot model using forward kinematics. The loss function is defined as the difference between the computed H1 joint positions and the corresponding SMPL joint positions. The optimization problem is defined as follows:

$$\begin{aligned} \min_{\boldsymbol{P}_{H1}} \quad & \|\text{FK}(\boldsymbol{P}_{H1}, \boldsymbol{R}_{root}, \boldsymbol{O}_{offset}) - \boldsymbol{P}_{SMPL}\|_2^2 + \lambda\|\boldsymbol{P}_{H1}\|_2^2, \\ \text{s.t.} \quad & \boldsymbol{P}_{lower} \leq \boldsymbol{P}_{H1} \leq \boldsymbol{P}_{upper}, \\ & \|\boldsymbol{P}_{H1}\|_2^2 \leq 4. \end{aligned} \quad (28)$$

The $\ell_2$ norm penalty ensures smoother values and prevents $\boldsymbol{P}_{H1}$ from becoming excessively large during optimization. Large control inputs could be impractical and could even damage the robot hardware. **1737 examples have been generated for this task via IPOPT**.

## H. Baseline Settings

### H.1. DC3

DC3 (Deep Constraint Completion and Correction) is a neural network-based constrained optimization solver. Unlike direct prediction of solutions via neural networks, DC3 incorporates two key components: an *equality completion* operator $\varphi_{\boldsymbol{x}}$ and an *inequality correction* operator $\rho_{\boldsymbol{x}}$. These mechanisms significantly improve the feasibility of the obtained solutions.

Moreover, DC3 adopts a self-supervised learning paradigm. Instead of requiring optimal solutions as supervision, it constructs its loss function as follows:

$$\ell_{\text{soft}}(\hat{\boldsymbol{y}}) = f(\hat{\boldsymbol{y}}; \boldsymbol{x}) + \lambda_g \cdot \|\text{ReLU}(g(\hat{\boldsymbol{y}}; \boldsymbol{x}))\|_2^2 + \lambda_h \cdot \|h(\hat{\boldsymbol{y}}; \boldsymbol{x})\|_2^2,$$

where $f(\cdot; \boldsymbol{x})$ denotes the objective function, while $g(\cdot; \boldsymbol{x})$ and $h(\cdot; \boldsymbol{x})$ represent inequality and equality constraints, respectively. The $\text{ReLU}$-based term penalizes constraint violations.

The training and sampling procedures of DC3 are outlined in Algorithm 2, adapted from (Donti et al., 2021). In our experiments, we fix $\lambda_g = \lambda_h = 5$. In this work, we define $N_\theta$ as a three-layer neural network comprising two hidden layers (each with 512 neurons, ReLU activation, batch normalization, and dropout with $p = 0.2$).

### H.2. MLP

In our experiments, we include an MLP baseline that shares the **identical network architecture** with DC3. However, the MLP differs in two key aspects: 1.) It only employs equality completion $\phi_{\boldsymbol{x}}$ and omits inequality correction $\rho_{\boldsymbol{x}}$. 2.) During training, it directly minimizes the MSE loss between predictions and ground-truth solutions:

$$\ell_{\text{mse}} = \|\widetilde{\boldsymbol{y}} - \boldsymbol{y}^\star\|_2^2.$$

## H.3. Model-Based Diffusion

We attempt to adopt the model-based diffusion method proposed in (Pan et al., 2024) as a baseline for this work. Since practical application scenarios may not strictly satisfy the conditions specified in the original paper, we adapt the method with specific modifications. Concretely, when calculating the probability score for each sample, we compute it as follows:

$$p_i = \mathrm{P}(\boldsymbol{y}_i|\boldsymbol{x}) := f(\boldsymbol{y}_i; \boldsymbol{x}) + \lambda_h \|h(\boldsymbol{y}_i; \boldsymbol{x})\|_2 + \lambda_g \|\mathrm{ReLU}(g(\boldsymbol{y}_i; \boldsymbol{x}))\|_2. \tag{29}$$

Here, $\boldsymbol{y}_i$ represents the $i$-th sample within the complete collection of samples in one diffuse step. The subsequent algorithmic steps remain consistent with Algorithm 1 in (Pan et al., 2024). For all experiments, the number of samples is set to 256, the number of diffusion steps is set to 100, and $\lambda_h = \lambda_g = 10$. The specific process of the model-based diffusion with completion is outlined in Algorithm 3, where "Completion" denotes the task-specific completion procedure.

## H.4. RectFlow

RectFlow (Liang & Chen, 2024) is an ODE-based conditional generative model for learning an input-conditioned solution distribution. Instead of predicting a single point estimate, RectFlow learns a continuous transport dynamics that maps a simple prior distribution to a solution distribution conditioned on the problem instance.

**Training** RectFlow assumes access to a fixed dataset of paired instances and solutions $\{(\boldsymbol{x}_i, \boldsymbol{y}_i)\}_{i=1}^N$, where each $\boldsymbol{y}_i$ is a feasible/near-optimal solution obtained offline (e.g., from a numerical solver or a dataset release). Given a prior $q(\boldsymbol{y}_0)$, RectFlow samples $\boldsymbol{y}_0 \sim q$, draws a time $t \sim \mathrm{Uniform}[0, 1]$, and forms the linear interpolation

$$\boldsymbol{y}_t = (1 - t)\boldsymbol{y}_0 + t\boldsymbol{y}_1, \quad \text{with } \boldsymbol{y}_1 = \boldsymbol{y}_i \text{ from the dataset.}$$

It then trains a neural vector field $v_\theta(\boldsymbol{y}, t, \boldsymbol{x})$ via the following objective:

$$\ell_{\mathrm{rf}}(\theta) = \mathbb{E}_{(\boldsymbol{x}, \boldsymbol{y}_1), \boldsymbol{y}_0, t}\left[\left\|v_\theta(\boldsymbol{y}_t, t, \boldsymbol{x}) - (\boldsymbol{y}_1 - \boldsymbol{y}_0)\right\|_2^2\right].$$

Intuitively, the target $(\boldsymbol{y}_1 - \boldsymbol{y}_0)$ represents the rectified displacement from noise to the data endpoint under a straight-line coupling.

**Generation** Given a new instance $\boldsymbol{x}$, RectFlow generates solutions by sampling $\boldsymbol{y}_0 \sim q(\boldsymbol{y}_0)$ and integrating the ODE

$$\frac{d\boldsymbol{y}_t}{dt} = v_\theta(\boldsymbol{y}_t, t, \boldsymbol{x}), \quad t \in [0, 1],$$

to obtain $\boldsymbol{y}_1$. In practice, a simple Euler solver with $K$ steps is used:

$$\boldsymbol{y}_{(k+1)/K} = \boldsymbol{y}_{k/K} + \frac{1}{K} v_\theta\left(\boldsymbol{y}_{k/K}, k/K, \boldsymbol{x}\right), \quad k = 0, \ldots, K - 1.$$

Following common practice in conditional generative solvers, RectFlow can also employ *solution selection*.

RectFlow shares a nearly identical training paradrigm with the supervised stage of DiOpt. Let $\boldsymbol{x}$ denote a problem instance, $\boldsymbol{y}_1$ the paired offline solution, $\boldsymbol{y}_0 \sim q(\boldsymbol{y}_0)$ a prior sample, and $t \sim \mathrm{Uniform}[0, 1]$ the time step. Both methods train a conditional generator on a fixed dataset $\{(\boldsymbol{x}_i, \boldsymbol{y}_i)\}_{i=1}^N$ by sampling interpolation points $\boldsymbol{y}_t = (1 - t)\boldsymbol{y}_0 + t\boldsymbol{y}_1$ and regressing a time-conditioned target representing the transport direction. The divergence lies solely in the parameterization: RectFlow learns a velocity field via flow matching, whereas DiOpt learns a noise predictor. Thus, DiOpt with only supervised stage effectively functions as a diffusion-based instantiation of the RectFlow.

---
**Algorithm 2** Deep Constraint Completion and Correction (DC3)

---
1: **Assume:** Equality completion procedure $\varphi_{\boldsymbol{x}} : \mathbb{R}^{d_{\boldsymbol{y}} - d_{\text{neq}}} \to \mathbb{R}^{d_{\text{neq}}}$
2: **procedure** TRAIN $(X)$
3: Initialize neural network $N_\theta : \mathbb{R}^{d_{\boldsymbol{x}}} \to \mathbb{R}^{d_{\boldsymbol{y}} - d_{\text{neq}}}$
4: **while** not converged **do**
5:    **for** $\boldsymbol{x} \in X$ **do**
6:       Compute partial variables: $\boldsymbol{z} = N_\theta(\boldsymbol{x})$
7:       Complete to full variables: $\tilde{\boldsymbol{y}} = \begin{bmatrix} \boldsymbol{z}^T & \varphi_{\boldsymbol{x}}(\boldsymbol{z})^T \end{bmatrix}^T \in \mathbb{R}^{d_{\boldsymbol{x}}}$
8:       Correct to feasible (or approx. feasible) solution: $\hat{\boldsymbol{y}} = \rho_{\boldsymbol{x}}^{(train)}(\tilde{\boldsymbol{y}})$
9:       Compute constraint-regularized loss: $\ell_{\text{soft}}(\hat{\boldsymbol{y}})$
10:      Update $\theta$ using $\nabla_\theta \ell_{\text{soft}}(\hat{\boldsymbol{y}})$
11:    **end for**
12: **end while**
13: **end procedure**
14: **procedure** TEST $(\boldsymbol{x}, N_\theta)$
15: Compute partial variables: $\boldsymbol{z} = N_\theta(\boldsymbol{x})$
16: Complete to full variables: $\tilde{\boldsymbol{y}} = \begin{bmatrix} \boldsymbol{z}^T & \varphi_{\boldsymbol{x}}(\boldsymbol{z})^T \end{bmatrix}^T$
17: Correct to feasible solution: $\hat{\boldsymbol{y}} = \rho_{\boldsymbol{x}}^{(test)}(\tilde{\boldsymbol{y}})$
18: **Return** $\hat{\boldsymbol{y}}$
19: **end procedure**

---

---
**Algorithm 3** Model-based Diffusion with completion

---
**Input:** $\boldsymbol{z}^{(N)} \sim \mathcal{N}(0, I)$, Condition Parameter $\boldsymbol{x}$, Number of Diffusion Steps $N$, Number of Samples $d$.
**for** $i = N$ **to** $1$ **do**
   Sample $d$ samples $\mathcal{Z}^{(i)} = [\boldsymbol{z}_1^i, ..., \boldsymbol{z}_d^i] \overset{i.i.d}{\sim} \mathcal{N}\left(\frac{\boldsymbol{z}^{(i)}}{\sqrt{\alpha_{i-1}}}, \left(\frac{1}{\bar{\alpha}_{i-1}} - 1\right) I\right)$
   Get completion: $\mathcal{Y}^{(i)} = [\boldsymbol{y}_1^i, ..., \boldsymbol{y}_d^i] = \text{Completion}(\mathcal{Z}^{(i)}; \boldsymbol{x})$
   Calculate probability score: $p_j = P(\boldsymbol{y}_j^i | \boldsymbol{x})$
   Estimate New Center: $\boldsymbol{z}^{(i-1)} = \frac{\sum_{j=1}^d p_j \boldsymbol{z}_j^i}{\sum_{j=1}^d p_j}$
**end for**
Complete partial solution: $\boldsymbol{y}^{(0)} = \text{Completion}(\boldsymbol{z}^{(0)}; \boldsymbol{x})$
**Return** Optimized solution $\boldsymbol{y}^{(0)}$

---

