# OpenReview forum: "Diffusion-based learning framework for Constrained Nonconvex Optimization with Weighted Bootstrapped Refinement"
_ICML.cc/2026/Conference — ICML 2026 regular_

### Official Review · Reviewer_4SCn · 2026-03-10

**Soundness:** 2
**Presentation:** 3
**Significance:** 3
**Originality:** 3
**Overall Recommendation:** 4
**Confidence:** 2

**Summary:**

This manuscript studies diffusion-based learning for constrained nonconvex optimization.
Facing the challenge of generating feasible and high-quality solutions, the manuscript proposes a two-stage framework, DiOpt, which first warm-starts the diffusion solver with supervised training and then refines it through a bootstrapped weighted refinement. The second stage is further augmented with several constraint-aware designs, including reset and look-up table,  among others. Experiments on multiple optimization tasks show that the proposed method achieves strong performance and improves the trade-off between feasibility and optimality over prior approaches.

**Compliance With Llm Reviewing Policy:**

Affirmed.

**Final Justification:**

Overall, my concerns are fully resolved. I would like to keep my positive score.

**Key Questions For Authors:**

See Weaknesses.

**Limitations:**

yes

**Strengths And Weaknesses:**

# Strengths
1. The paper provides a clear discussion of the limitations of existing methods, which makes the motivation well grounded.

2. The work includes detailed theoretical analysis that helps support the main intuition of the method.

3. The proposed two-stage training framework is well motivated, and the accompanying designs are tailored to constrained optimization.

4. The method demonstrates strong performance across the evaluated tasks.

# Weaknesses
1. The ablations are still limited. While the appendix includes a number of ablations, many of them focus primarily on hyperparameter choices rather than demonstrating the necessity of the key design components.
As a result, it remains unclear whether some of the design choices in the second-stage refinement are truly essential, such as the look-up table.

2. Since the second-stage refinement is explicitly motivated by QVPO-style diffusion training,  even if a direct comparison with RL-based strategies may be difficult due to architectural differences, the paper should at least compare against a simpler baseline in the same spirit, such as a generic weighted self-bootstrapping variant. Without such a baseline, it is difficult to see whether the gains mainly come from the general weighted post-training paradigm.

---

> ### Author Rebuttal · Authors · 2026-03-31
>
> We thank Reviewer 4SCn for engaging so deeply with our work. The thoughtful questions raised have enabled us to articulate our contributions more clearly and address potential ambiguities.
>
>
> > **Q1**: The ablations are still limited. While the appendix includes a number of ablations, many of them focus primarily on hyperparameter choices rather than demonstrating the necessity of the key design components. As a result, it remains unclear whether some of the design choices in the second-stage refinement are truly essential, such as the look-up table.
>
>
> **A1**: Thank you very much for your feedback. Here we further provide an ablation study on QPSR.
>
>
> | Configuration | Gap $\downarrow$   | Violation $\downarrow$ |
> |---------------|---------|----------|
> | w.  look-up table          | **6.69%**   | **0.008**    |
> | w.o.    look-up table      | 112.8%  | 0.120    |
>
> According to the table, the use of a lookup table is essential. Due to the randomness inherent in sampling, we cannot guarantee that every sample will be better than the previous one. Without a lookup table, the model would be directly affected by inferior solutions caused by sampling stochasticity, making it impossible to ensure consistent improvement of the model.
>
>
> > **Q2**: Since the second-stage refinement is explicitly motivated by QVPO-style diffusion training, even if a direct comparison with RL-based strategies may be difficult due to architectural differences, the paper should at least compare against a simpler baseline in the same spirit, such as a generic weighted self-bootstrapping variant. Without such a baseline, it is difficult to see whether the gains mainly come from the general weighted post-training paradigm.
>
>
> **A2**: Thanks for your valuable suggestion. To illustrate the effectiveness of our weighting mechanism, we conducted an ablation study with a naive penalty‑based weighting:
>
> $$
> w(y;x) = \exp\left(-f(y;x) - 10\cdot \sum_i \max(g_i(y;x),0)\right)
> $$
>
> We take QPSR as an example and the results are shown below. As observed, the penalty‑based weighting leads to larger violations and gap.
> | Weight Type | Gap $\downarrow$   | Violation $\downarrow$ |
> | ----------- | ----- | --------- |
> | Ours    | **6.69%** | **0.008**     |
> | Penalty（additional baseline）    | 8.17% | 0.03      |

---

> > ### Author Rebuttal · Reviewer_4SCn · 2026-04-01
> >
> > The rebuttal is helpful. The authors add new experiments, including the look-up-table ablation and an additional weighting baseline. These additions strengthen the empirical support for the method.

---

> > > ### Author Response · Authors · 2026-04-01
> > >
> > > Thank you for your positive feedback and for acknowledging the revisions made in the rebuttal.
> > >
> > > We are glad to hear that you found the new experiments helpful in strengthening the empirical support for our method.
> > >
> > > Thank you again for your time and valuable feedback throughout the review process.

---

### Official Review · Reviewer_bXyp · 2026-03-11

**Soundness:** 3
**Presentation:** 3
**Significance:** 2
**Originality:** 2
**Overall Recommendation:** 4
**Confidence:** 3

**Summary:**

This paper studies diffusion-based learning for constrained nonconvex optimization and attributes the low feasibility of prior supervised diffusion solvers to a mismatch between the learned solution distribution and the feasible region. It proposes a weighted bootstrapping framework that progressively shifts sampling toward near-optimal feasible solutions and further improves inference through candidate selection.

**Compliance With Llm Reviewing Policy:**

Affirmed.

**Final Justification:**

I appreciate the authors’ further explanation of the weighting/refinement design and the additional appendix results. The rebuttal is helpful and addresses my concerns to a meaningful extent, while leaving some room for further strengthening in the final version, particularly regarding empirical positioning and comparison. I therefore keep my score unchanged.

**Key Questions For Authors:**

1. It would be helpful to better justify why the weighting strategy and bootstrapped refinement are principled and necessary, rather than mainly heuristic design choices.
2. How does the proposed method compare with more recent constrained diffusion approaches, in both effectiveness and computational efficiency across tasks?

**Strengths And Weaknesses:**

Strengths:
1. The paper addresses an important problem and gives a clear explanation of why supervised diffusion often fails under hard constraints, especially in high-dimensional settings.
2. The proposed method is relatively mature and targets both feasibility and objective quality rather than optimizing only one side.

Weaknesses
1. The current theory mainly explains the limitation of supervised diffusion. Meanwhile, the weighting design and refinement mechanism remain somewhat heuristic.
2. The paper’s positioning is still not strong enough against the latest constrained diffusion methods, especially newer 2025 methods. It would also benefit from more systematic efficiency analysis across tasks to justify its practical advantage.

---

> ### Author Rebuttal · Authors · 2026-03-31
>
> We are sincerely grateful to Reviewer bXyp for the meticulous review and the favorable comments regarding the significance of our findings. Your appreciation of our work is both encouraging and validating.
>
>
> > **Q1**: It would be helpful to better justify why the weighting strategy and bootstrapped refinement are principled and necessary, rather than mainly heuristic design choices.
>
> **A1**: Regarding the design of the weighting, we provide the underlying rationale in Equation (9):
> $$
> p(y;x)\sim \mathbb{I}_{C(x)}(y)\exp(-\beta f(y;x)) \tag{9}
> $$
> As established in [R1], Equation (9) converges to the optimum as $\beta \to \infty$. This theoretical guarantee justifies our weighting design and refinement mechanism. Conceptually, our approach parallels the exact penalty method [R2, p. 421] which assigns higher score to feasible samples and lower scores to infeasible ones, ensuring applicability to both scenarios.
>
> [R1]: Pan, Chaoyi, et al. "Model-based diffusion for trajectory optimization." Advances in Neural Information Processing Systems 37 (2024): 57914-57943.
> [R2]: Luenberger, David G., and Yinyu Ye. Linear and Nonlinear Programming. 4th ed., Springer, 2016.
>
> > **Q2**: How does the proposed method compare with more recent constrained diffusion approaches, in both effectiveness and computational efficiency across tasks?
>
> **A2:** DiffuSolve [R1], a recent diffusion-based solver that remains **unopen-sourced**, adopts similiar supervised learning paradigm as RectFlow [R2], **lacking the self-supervised bootstrapping** mechanism for constraint handling. As evidenced by RectFlow’s **0% feasibility** on QPSR, CQP, and ACOPF118 in Table 2, such methods may fail to ensure constraint satisfaction as well. In contrast, DiOpt achieves 84.33% feasibility on ACOPF118 with inference time comparable to theirs.
>
> [R1]: Li, Anjian, et al. "DiffuSolve: Diffusion-based Solver for Non-convex Trajectory Optimization." 7th Annual Learning for Dynamics\& Control Conference. PMLR, 2025.
> [R2]: Liang E, Chen M. Generative learning for solving non-convex problem with multi-valued input-solution mapping[C]//The Twelfth International Conference on Learning Representations. 2024.

---

> > ### Author Rebuttal · Reviewer_bXyp · 2026-04-01
> >
> > Thank you for the rebuttal. I appreciate the clarifications on both the weighting/refinement design and the comparison with recent constrained diffusion methods. These concerns are addressed to some extent, but not yet fully. Some design choices still appear somewhat heuristic, and the empirical positioning/computational comparison remains limited. Therefore, I will keep my score unchanged.

---

> > > ### Author Response · Authors · 2026-04-01
> > >
> > > We appreciate the reviewer's careful reading and thoughtful comments. However, we would like to clarify that our method **is not heuristic** and the weighting design is derived from verified constrained optimization techniques, specifically [R1] and the Exact Penalty Term, [R2, p. 421]. Besides, we highlight that **Appendices E and F** provide **comprehensive ablation studies** and **detailed computational analyses** across all hyperparameters, while **Table 2** reports inference time comparisons across all benchmarks. Training time data is further provided in our response to **Reviewer BsVv in A2**. Furthermore, Appendices A, B, and C establish the **theoretical foundations** regarding feasibility bounds, multiple sampling limitations, and reset mechanism analysis. These extensive validations demonstrate that our approach is built on solid methodological foundations and thoroughly benchmarked.
> > >
> > > [R1]: Pan, Chaoyi, et al. "Model-based diffusion for trajectory optimization." Advances in Neural Information Processing Systems 37 (2024): 57914-57943.
> > >
> > > [R2]: Luenberger, David G., and Yinyu Ye. Linear and Nonlinear Programming. 4th ed., Springer, 2016.

---

### Official Review · Reviewer_BsVv · 2026-03-11

**Soundness:** 3
**Presentation:** 3
**Significance:** 2
**Originality:** 3
**Overall Recommendation:** 4
**Confidence:** 3

**Summary:**

This works identifies a key challenge for existing diffusion-based solvers that they fail to generate feasible solutions. Such challenges are theoretically justified through low probability mass of their generated solutions in the feasible region. To resolve this issue, the authors propose DiOPt to train the diffusion solver by two phases: an initial warm-start phase followed by a bootstrapping training phase, and further propose a solution selection technique in inference for better optimality. Evaluations on diverse nonconvex constrained optimization problem suites demonstrate the superiority of DiOpt in both optimality and constraint satisfaction.

**Compliance With Llm Reviewing Policy:**

Affirmed.

**Final Justification:**

The author rebuttal has addressed most of my concerns. My evaluation on its soundness has improved and elevated this work above the acceptance bar. I have increased my score following this change.

**Key Questions For Authors:**

- It seems that the weighting schema in (11) introduces some discontinuity: for $y \in C(x)$, the minimum value is $\exp(0)=1$, while for $y \notin C(x)$, the maximum value is 0. Is this done on purpose? Some explanation may be welcome here.
- Regarding the time cost of learning/non-learning-based optimization algorithms, a more fair comparison will be including both the training and inference time. Nevertheless, such results seem missing in current version. Given the large number of training epochs and diffusion steps in Table 5, some clarification should be necessary here.
- The authors may also need to clarify their inference setting: while the training stage may naturally involve GPUs, the inference stage should only be conducted on CPU for fair comparison on the time cost.
- It also seems that the authors do not include details of their training data in current draft. One might wonder how these training data are obtained/generated, and whether they are similar to benchmark data. Some clarification is also necessary here to avoid “training on the testing set/distribution”.
- Typos & Formatting issues
  - Algorithm 1, Line 333: should $t$ be $n$ instead?
  - The paragraphs in Section 3 are not separated properly: the “Diffusion Models” paragraph gets mixed in the main text

**Limitations:**

Yes

**Strengths And Weaknesses:**

Strength:
- (Presentation) The proposed method is generally easy to follow
- (Originality) The idea in this work is novel to the best of my knowledge

Weaknesses:
- (Soundness) Empirical results and some details can be improved to better support the proposed method
- (Significance) The contribution of this work seems to be restricted to diffusion solvers

---

> ### Author Rebuttal · Authors · 2026-03-31
>
> We are deeply grateful to Reviewer BsVv for the comprehensive and constructive feedback. The detailed suggestions have helped us clarify key concepts and strengthen the technical rigor of our work.
>
> > **Q1**. It seems that the weighting schema in (11) introduces some discontinuity: for $y \in C(x)$, the minimum value is $\exp(0) = 1$, while for $y \notin C(x)$, the maximum value is 0. Is this done on purpose? Some explanation may be welcome here.
>
> **A1**: Here we would like to point out a minor issue. Since the optimal value is strictly smaller than the candidate value, $\exp(f^{*}(x) - f(y;x)) \in (0,1)$. Hence, no discontinuity is present. Furthermore, we assign negative weights to infeasible cases to encourage the model to prioritize feasible points, thereby enhancing feasibility.
>
> > **Q2**: Regarding the time cost of learning/non-learning-based optimization algorithms, a more fair comparison will be including both the training and inference time. Nevertheless, such results seem missing in current version. Given the large number of training epochs and diffusion steps in Table 5, some clarification should be necessary here.
>
> **A2**: We clarify that inference speed is important for real-time applications like power grid control. This is the core motivation for learning-based optimization methods: train once, infer rapidly at deployment. Therefore, Table 5 reports the inference time for DiOpt, DC3, and RectFlow. We acknowledge that training time is also an important metric, and we will include it in the revised paper. Below we list the training times.
>
> | Method | QP | CQP | QPSR | acopf57 | acopf118 | Retargeting |
> |:------:|:--:|:---:|:----:|:-------:|:--------:|:-----------:|
> | DC3 | 0.67h | 0.67h | 0.67h | 2.52h | 13.05h | 0.33h |
> | DiOpt | 3.75h | 3.75h | 3.75h | 9.48h | 33.12h | 1.02h |
>
> As noted in the Conclusion, DiOpt has longer training times due to its self-supervised objective and equality completion, but this brings better optimality gap and feasibility during inference. Unlike traditional solvers, DiOpt does not need retraining for new instances, making this one-time cost worthwhile.
>
> > **Q3**. The authors may also need to clarify their inference setting: while the training stage may naturally involve GPUs, the inference stage should only be conducted on CPU for fair comparison on the time cost.
> >
> **A3**: Thank you for your feedback. On this point, it is a common practice in the field to report GPU runtime for neural solvers, as neural networks are inherently capable of efficiently leveraging the powerful parallel computing capabilities of GPUs. This is also the case for learning-based optimization methods such as DC3 [R1] and RectFlow [R2]. Therefore, in the table, we report the GPU inference time for both RectFlow, DC3 and our DiOpt.
> [R1]: Donti, Priya L., David Rolnick, and J Zico Kolter. "DC3: A Learning Method for Optimization with Hard Constraints." International Conference on Learning Representations, 2021
> [R2]: Liang, Enming, and Minghua Chen. "Generative learning for solving non-convex problem with multi-valued input-solution mapping." The Twelfth International Conference on Learning Representations. 2024.
>
> > **Q4**: It also seems that the authors do not include details of their training data in current draft. One might wonder how these training data are obtained/generated, and whether they are similar to benchmark data. Some clarification is also necessary here to avoid “training on the testing set/distribution”.
>
>
> **A4**: Thank you for your patience in reading. Regarding this issue, we provided a detailed description of our benchmarks, including how they were constructed and generated, in Appendix G.
>
> > **Q5**: Typos & Formatting issues
> >   - Algorithm 1, Line 333: should $t$ be $n$ instead?
> >   - The paragraphs in Section 3 are not separated properly: the “Diffusion Models” paragraph gets mixed in the main text
>
> - In line 333, there is a table, and we did not find the two symbols you mentioned. As to the Algorithm 1, we also did not find them elsewhere. We guess you may refer to $K_t$ and it denotes the number of training samples without any mistake.
> - Regarding “Diffuison Models,” this was an oversight. We will correct it in our final version.

---

> > ### Author Rebuttal · Reviewer_BsVv · 2026-04-01
> >
> > I would like to first thank the authors for their detailed responses. Most of my previous concerns have been addressed. One minor concerns remains regarding the notation in Algorithm 1: should the $t$ around line 231-233 be $n$ or some other indices? I am well aware of $K_t$, while its subscript $t$ still makes me a bit confused.

---

> > > ### Author Response · Authors · 2026-04-01
> > >
> > > Thank you for your careful reading. As you mentioned, the variable $t$  at line 231 of Algorithm 1 is a typo and should be corrected to $n$  (representing the current epoch). We will correct it in our final version.
> > >
> > > We sincerely appreciate your patience in reviewing our manuscript and your recognition of our work. We are delighted that our response has resolved your confusion. Thank you again for your time and valuable feedback throughout the review process.

---

### Decision · Program_Chairs · 2026-04-30

**Decision:**

Accept (regular)

**Comment:**

This paper addresses the core challenge that existing diffusion-based solvers fail to generate feasible solutions, and proposes DiOpt—a weighted bootstrapping optimization method built on a two-stage diffusion learning framework. As the first work that successfully applies diffusion solvers to constrained nonconvex optimization, this paper clearly clarifies the core limitations of existing methods and designs a supervised warm-start plus bootstrapping training architecture. Extensive experiments on diverse nonconvex tasks verify its superior performance over state-of-the-art learning-based optimizers.

Strengths
The paper accurately identifies and theoretically analyzes the inherent defects of existing diffusion-based optimization methods, with solid and convincing motivation.
The two-stage DiOpt framework is novel and fills the gap in the application of diffusion models to constrained nonconvex optimization.
The weighting mechanism is derived from the exact penalty function method with solid theoretical support; key designs are validated through targeted ablation experiments.
The method significantly outperforms benchmark approaches in both optimality gap and constraint satisfaction, and the authors’ rebuttal effectively addresses most reviewers’ concerns.

Weaknesses

The final camera-ready version should supplement and emphasize the one-time training cost and its practical application value.
The theoretical justification of weighted bootstrapping should be further strengthened to reduce the heuristic impression.
Overall, this work makes a pioneering contribution to diffusion-based constrained nonconvex optimization with solid technicality, sufficient experiments and clear academic merit. It is recommended for acceptance as a poster presentation, and the authors are expected to complete the above minor revisions in the final version.